# Macrophage migration inhibitory factor is required for NLRP3 inflammasome activation

Tali Lang[1,14], Jacinta P.W. Lee[1], Kirstin Elgass[2], Anita A. Pinar[1,15], Michelle D. Tate[3,4], Elizabeth H. Aitken[5], Huapeng Fan[1], Sarah J. Creed[2], Nadia S. Deen[1], Daouda A.K. Traore [6,7], Ivo Mueller [8,9,10], Danielle Stanisic[11,16], Francesca S. Baiwog[12], Colin Skene[13], Matthew C.J. Wilce[6], Ashley Mansell[3,4], Eric F. Morand [1] & James Harris[1]

Macrophage migration inhibitory factor (MIF) exerts multiple effects on immune cells, as well as having functions outside the immune system. MIF can promote inflammation through the induction of other cytokines, including TNF, IL-6, and IL-1 family cytokines. Here, we show that inhibition of MIF regulates the release of IL-1α, IL-1β, and IL-18, not by affecting transcription or translation of these cytokines, but via activation of the NLRP3 inflammasome. MIF is required for the interaction between NLRP3 and the intermediate filament protein vimentin, which is critical for NLRP3 activation. Further, we demonstrate that MIF interacts with NLRP3, indicating a role for MIF in inflammasome activation independent of its role as a cytokine. These data advance our understanding of how MIF regulates inflammation and identify it as a factor critical for NLRP3 inflammasome activation.

[1] Rheumatology Research Group, Centre for Inflammatory Diseases, School of Clinical Sciences at Monash Health, Faculty of Medicine, Nursing & Health Sciences, Monash University, Clayton, VIC 3168, Australia. [2] Monash Micro Imaging, Hudson Institute of Medical Research, Clayton, VIC 3168, Australia. [3] Centre for Innate Immunity and Infectious Disease, Hudson Institute of Medical Research, Clayton, VIC 3168, Australia. [4] Department of Molecular and Translational Sciences, Monash University, Clayton, VIC 3168, Australia. [5] Department of Medicine, Peter Doherty Institute, University of Melbourne, Parkville, VIC 3800, Australia. [6] Department of Biochemistry and Molecular Biology, Biomedicine Discovery Institute, Monash University, Clayton, VIC 3000, Australia. [7] Faculte des Sciences et Techniques, Universite des Sciences Techniques et Technologiques de Bamako (USTTB), Bamako, Mali. [8] Population Health and Immunity Division, Walter and Eliza Hall Institute, Parkville, 3052 VIC, Australia. [9] Department of Medical Biology, University of Melbourne, Parkville, 3052 VIC, Australia. [10] Malaria: Parasites and Hosts Unit, Institut Pasteur, 75015 Paris, France. [11] Infection and Immunity Division, Walter and Eliza Hall, Institute of Medical Research, Parkville, VIC 3000, Australia. [12] Papua New Guinea Institute of Medical Research, Madang, Papua New Guinea. [13] Department of Chemistry, University of Melbourne, Parkville, VIC 3000, Australia. [14] Present address: The Szalmuk Family Department of Medical Oncology, Cabrini Institute Malvern, Malvern, VIC 3144, Australia. [15] Present address: Department of Pharmacology, Faculty of Medicine, Nursing and Health Sciences, Monash University, 9 Ancora Imparo Way, Clayton, VIC 3800, Australia. [16] Present address: Institute for Glycomics, Griffith University, Southport, QLD 4215, Australia. Correspondence and requests for materials should be addressed to J.H. (email: jim.harris@monash.edu)

Macrophage migration inhibitory factor (MIF) is a pluripotent protein with a broad range of immunomodulatory functions. It is produced and stored within cytoplasmic vesicles in many cell types including macrophages, T cells, and dendritic cells, enabling rapid release in response to a range of stimuli[1]. MIF has topological homology to three bacterial enzymes, oxalocrotonate tautomerase, 5-carboxymethyl-2-hydroxymuconate isomerase, and chorismate mutase[2–6], and has an enzymatic activity as a D-dopachrome tautomerase, phenylpyruvate tautomerase, and a thiol-protein oxidoreductase[2,7,8]. However, the relationship between the catalytic activity and the biological functions of MIF is unclear. MIF is an important regulator of protective responses to intracellular pathogens, including Gram-negative bacteria, fungal pathogens, viruses, and protozoa[9–12]. However, MIF is also linked to the pathogenesis of inflammatory and autoimmune disorders, including sepsis, rheumatoid arthritis, and systemic lupus erythematosus[13,14]. Amongst its many effects on immune cell function, MIF has been shown to modulate the production and secretion of proinflammatory cytokines, including tumor necrosis factor (TNF), interleukin-6 (IL-6), interferon-γ, and interleukin-1β (IL-1β)[1,13]. In particular, MIF can regulate TNF and IL-6 through effects on the expression of Toll-like receptor 4 (TLR4), p53, ERK, mitogen-activated protein kinase (MAPK), c-Jun-N-terminal kinase, p38, and MAPK phosphatase-1[1,15]; however, the molecular mechanisms by which MIF regulates the secretion of IL-1β are not well understood.

IL-1α, IL-1β, and IL-18 have important functions in mediating innate and adaptive immunity. IL-1α and IL-1β bind the same receptor, IL-1R1, and share similar proinflammatory properties mainly through the induction of cyclooxygenase type-2, type 2 phospholipase A, and inducible nitric oxide synthase, leading to recruitment of myeloid cells, including neutrophils, to sites of inflammation[16]. Moreover, in concert with IL-23, IL-1α and IL-1β can drive the expression and secretion of IL-17 and IL-22 from T-helper type 17 (Th17) and γδ T cells[17]. Although structurally homologous to IL-1α and IL-1β, IL-18 binds a different receptor and has distinct functions. In particular, IL-18, in combination with IL-12, stimulates Th1-mediated responses, including the secretion of interferon-γ (IFN-γ) by T cells and natural killer cells[18]. Both IL-1β and IL-18 are initially produced as biologically inactive pro-forms that require cleavage into the mature cytokines. Typically, this processing is mediated by caspase-1, which is activated following the formation of an inflammasome.

Inflammasomes are multimeric scaffolding complexes that activate caspase-1[19]. Several inflammasome complexes have been identified, with most incorporating at least one adaptor protein such as an AIM2-like receptor (ALR), pyrin protein, or a nucleotide-binding domain, leucine-rich-repeat-containing protein (NOD-like receptor, NLR), such as NLRP1, NLRC4, or NLRP3[19–22]. This ALR or NLR engages apoptosis-associated speck-like protein containing a caspase activation and recruitment domain (ASC), which, in turn, recruits and activates caspase-1. Activation of the NLRP3 inflammasome is a two-step process. First, a priming signal, such as a TLR ligand, promotes transcription of pro-IL-1β and NLRP3 via nuclear factor-κB (NF-κB)-mediated signaling[23]. A second signal is then required for the formation of the NLRP3 inflammasome complex. Numerous stimuli induce NLRP3 inflammasome formation, including extracellular adenosine triphosphate (ATP), pore-forming toxins such as nigericin[24] and particulates such as uric acid crystals, silica, and alum[25–27], as well as bacterial, protozoan, and viral pathogens[28–30].

The molecular mechanisms involved in NLRP3 inflammasome assembly are incompletely understood, but studies have demonstrated roles for NIMA-related kinase-7 (NEK7) and the type III intermediate filament protein vimentin[31–34]. Both proteins have been shown to interact with NLRP3, suggesting direct roles in NLRP3 inflammasome assembly and/or signaling. Although IL-1 family cytokines have important functions in protective immunity to various pathogens, dysregulation of these cytokines is also associated with pathology in a number of diseases, including cryopyrin-associated periodic syndromes, autoinflammatory syndromes, gout, type II diabetes, and some cancers[35]. Thus, understanding how inflammasome activation is regulated is critical for the development of better treatment strategies. Significantly, one study has demonstrated that MIF is required for IL-1β release and neutrophil recruitment in a mouse model of monosodium urate (MSU) crystal-induced gout[36], suggesting a possible function of MIF in the regulation of IL-1 in the specific context of NLRP3 activation.

Here we describe a specific function of MIF as a regulator of the NLRP3 inflammasome complex in macrophages. Inhibition of MIF in macrophages and dendritic cells inhibits NLRP3-dependent secretion of IL-1β and IL-18 in vitro and in vivo. Moreover, we demonstrate that MIF is required for the NLRP3–vimentin interaction and that MIF interacts with NLRP3. Our data identify a function of MIF in regulating NLRP3 inflammasome assembly and activation.

## Results

**MIF is required for the release of IL-1 family cytokines.** To assess the effects of MIF on the release of IL-1 family cytokines, bone marrow-derived macrophages (BMDM) were isolated from both wild-type (WT) and $Mif^{-/-}$ mice. Cells were then treated with the TLR4 agonist lipopolysaccharide (LPS) alone or in combination with the NLRP3 inflammasome activator nigericin. In both WT and $Mif^{-/-}$ BMDM, IL-1α, IL-1β, and IL-18 were secreted in response to LPS and nigericin only (Fig. 1a–c). However, secretion of both cytokines was significantly lower in $Mif^{-/-}$ cells (Fig. 1a, b). Interestingly, secretion of TNF-α in response to LPS was unaffected in $Mif^{-/-}$ BMDM (Supplementary Figure 1A), although IL-6 secretion was significantly lower (Supplementary Figure 1B). Similar effects on levels of IL-1α and IL-1β production from $Mif^{-/-}$ bone marrow-derived dendritic cells (BMDCs) were observed following treatment with LPS and the NLRP3 activators nigericin or ATP (Supplementary Figure 1C and 1D).

We generated and screened a library of small molecule MIF antagonists, identifying COR123625 as a potent inhibitor of MIF that binds the molecule with high affinity (Supplementary Figure 2A and Supplementary Table 1). Initial structural validation studies confirmed that COR123625 bound within the enzymatic tautomerase site of MIF near the catalytic N-terminal proline-1 residue[37] (Supplementary Figure 2B), inhibiting tautomerase activity in a dose-dependent manner (Supplementary Figure 2C). We then assessed the effects of COR123625 on secretion of IL-1 family cytokines by macrophages. Consistent with our findings in $Mif^{-/-}$ BMDM, pre-treatment of immortalized BMDM with COR123625 had similar significant inhibitory effects on IL-1β secretion at concentrations of 25, 50 and 100 μM following activation with LPS and nigericin (Supplementary Figure 2D), but had no effect on TNF-α secretion in response to LPS (Supplementary Figure 2E). In primary WT BMDM COR123625 similarly inhibited the secretion of IL-1α, IL-1β, and IL-18 in response to LPS and nigericin (Fig. 1d–f), but had no effect on LPS-induced TNF-α secretion (Supplementary Figure 3A). Interestingly, in contrast to $Mif^{-/-}$ BMDM, COR123625 also had no effect on LPS-induced IL-6 secretion (Supplementary Figure 3B), possibly suggesting differences between chronic MIF deficiency versus acute MIF inhibition on IL-6 release by macrophages. COR123625 also inhibited IL-1β release in response to the TLR2 ligand Pam$_3$CSK$_4$

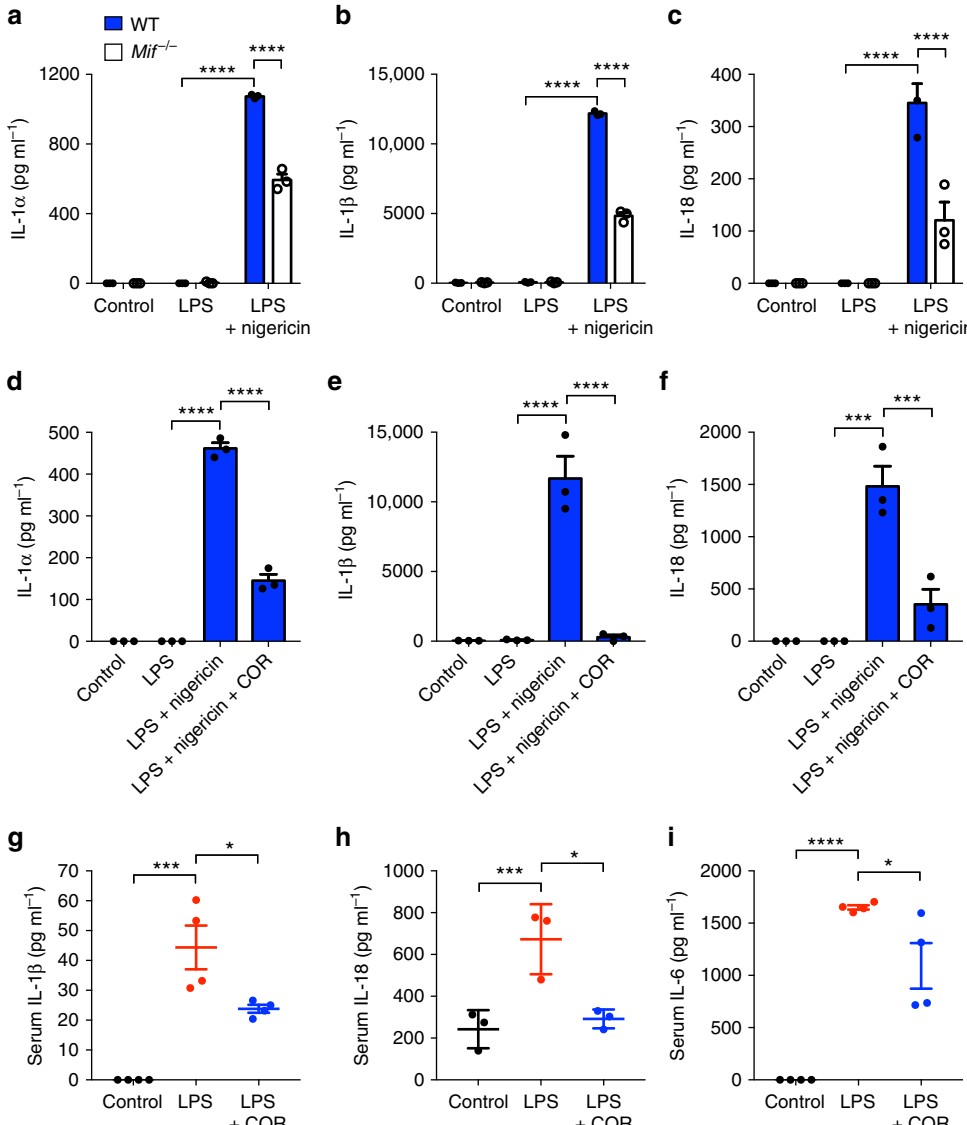

**Fig. 1** MIF is required for the release of IL-1 family cytokines. Primary murine WT and $Mif^{-/-}$ BMDMs were left untreated (control), primed with LPS (10 ng ml$^{-1}$), or primed with LPS followed by nigericin (5 µM) treatment for 1 h. Levels of **a** IL-1α, **b** IL-1β, and **c** IL-18 in cell culture supernatants were assessed by ELISA. Primary WT BMDMs were left untreated, primed with LPS alone (10 ng ml$^{-1}$), primed with LPS, and then treated with or without COR123625 (50 µM) for 2 h before the addition of nigericin (5 µM) for 1 h. Levels of **d** IL-1α, **e** IL-1β, and **f** IL-18 in cell culture supernatants were assessed by ELISA. C57BL6/J mice were injected intraperitoneally with vehicle control (saline), LPS alone (2 mg kg$^{-1}$) or COR123625 (20 mg kg$^{-1}$) in combination with LPS for 2 h. Serum levels of **g** IL-1β, **h** IL-18, and **i** IL-6 were measured by ELISA. Data are expressed as means ± SEM, $n = 3$–4 mice per group. $*P < 0.05$, $***P < 0.005$, $****P < 0.001$, one-way ANOVA with a correction provided by the Tukey's multiple comparisons test

with nigericin and the TLR7 ligand imiquimod with nigericin (Supplementary Figure 4A and 4B). Moreover, the commercially available MIF inhibitor 4-iodo-6-phenylpyrimidine (4-IPP) also inhibited IL-1β release by BMDM in response to LPS and nigericin (Supplementary Figure 4C). Similarly, 4-IPP and another MIF inhibitor, (S,R)-3-(4-Hydroxyphenyl)-4,5-dihydro-5-isoxazole acetic acid (ISO-1), inhibited both IL-1α and IL-1β release by human THP-1 monocytic cells in response to LPS and nigericin (Supplementary Figure 4D and 4E).

As both MIF and NLRP3 have been implicated in the pathogenesis of septic shock[38–40], we next investigated the effects of COR123625 on serum IL-1β and IL-18 in a mouse model of LPS-induced endotoxemia. Intraperitoneal injection of LPS resulted in significantly increased serum IL-1β and IL-18 compared to sham-treated mice, but this was abrogated by pre-treatment with COR123625 (Fig. 1g, h). In this in vivo setting, COR123625 also reduced serum IL-6

(Fig. 1i), possibly suggesting effects on cells other than macrophages, different signaling pathways, or knock-on effects of IL-1 inhibition in macrophages/DCs. However, LPS-induced elevation of serum IL-6 was not significantly inhibited in $Mif^{-/-}$ mice (Supplementary Figure 4F). Serum IL-1β, on the other hand, was significantly lower in $Mif^{-/-}$ mice, compared to WT mice (Supplementary Figure 4G). Moreover, co-injection of COR123625 into the LPS-treated $Mif^{-/-}$ mice had no further effect on levels of IL-6 or IL-1β (Supplementary Figure 4F and 4G). These data demonstrate that inhibition or deficiency of MIF preferentially inhibits IL-1 family cytokines both in vitro and in vivo.

**MIF is required for NLRP3-dependent IL-1β release.** We next looked at whether MIF is required for IL-1β release in response to other inflammasome stimuli. Following priming of both WT and $Mif^{-/-}$ BMDM with LPS, the AIM2 inflammasome was activated

via transfection of poly (deoxyadenylic-deoxythymidylic) (poly(dA: dT)) into cells. No difference was observed in AIM2-dependent IL-1β release between WT and $Mif^{-/-}$ cells (Fig. 2a). Similarly, inhibition of MIF with COR123625 had no effect on levels of secreted IL-1β in response to poly(dA:dT) (Fig. 2b). Moreover, COR123625 had no effect on flagellin-induced (NLRC4-dependent) IL-1β release (Fig. 2c). In addition, IL-1α release was similarly unaffected by COR123625 in these experiments (Supplementary Figure 5A–5C). These data suggest that the effects of MIF on IL-1 release may be specific to the NLRP3 inflammasome. To investigate

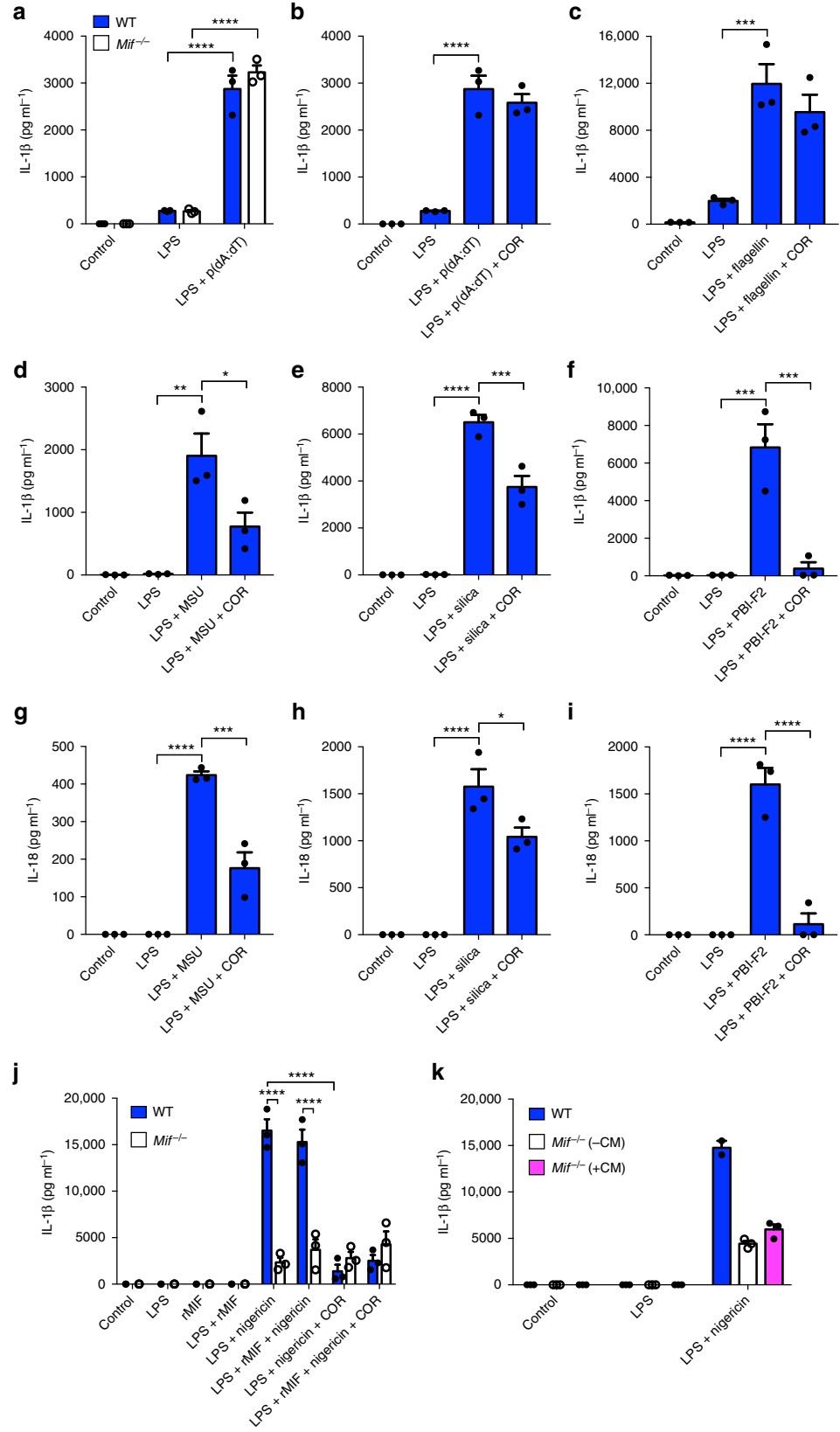

this further, we tested the effects of COR123625 on IL-1β release by BMDM in response to other known NLRP3 stimuli, including MSU crystals[26], silica[25], and the influenza A peptide, PB1-F2[30]. In each case, pre-treatment of cells with COR123625 significantly inhibited the release of both IL-1β and IL-18 (Fig. 2d–i). Similarly, IL-1β release by $Mif^{-/-}$ BMDM was significantly lower in response to MSU and silica compared to WT cells (Supplementary Figure 5D and 5E).

To determine whether exogenous MIF could reconstitute NLRP3-dependent IL-1β release, we treated both WT and $Mif^{-/-}$ BMDM with recombinant murine MIF (rMIF) and stimulated the cells with LPS and nigericin. While $Mif^{-/-}$ BMDM again demonstrated impaired IL-1β release in response to LPS and nigericin, the addition of rMIF had no effect on this (Fig. 2j). Moreover, while COR123625 inhibited NLRP3-dependent IL-1β release by WT BMDM, it had no further effect on $Mif^{-/-}$ cells (Fig. 2j), confirming that it does not have MIF-independent effects on IL-1β release in this setting. To test this further, we also stimulated $Mif^{-/-}$ cells with LPS and nigericin

after culturing the cells in MIF-containing conditioned medium from WT BMDM. This conditioned medium was taken from resting WT cells and MIF concentration was measured at 2428 pg ml$^{-1}$ by enzyme-linked immunosorbent assay (ELISA). Again, reconstituting MIF this way had no effect on NLRP3-dependent IL-1 release (Fig. 2k). Together, these data suggest that exogenous MIF does not facilitate NLRP3 activation, but that MIF instead acts in a cell-intrinsic, intracellular manner.

### ISO-1 inhibits IL-1β in response to *Plasmodium falciparum*.

Studies have highlighted a role for NLRP3-dependent IL-1β release in the immune response involved in the pathogenesis of malaria[28,29,41]. Of particular relevance, IL-1 is thought to play a pivotal role in low birth weight due to placental malaria[42,43]. Moreover, placental malaria is also associated with increased placental MIF[44]. Thus, to test whether MIF inhibition might reduce IL-1β release in the context of malaria, we treated peripheral blood mononuclear cells (PBMCs) isolated from pregnant women in

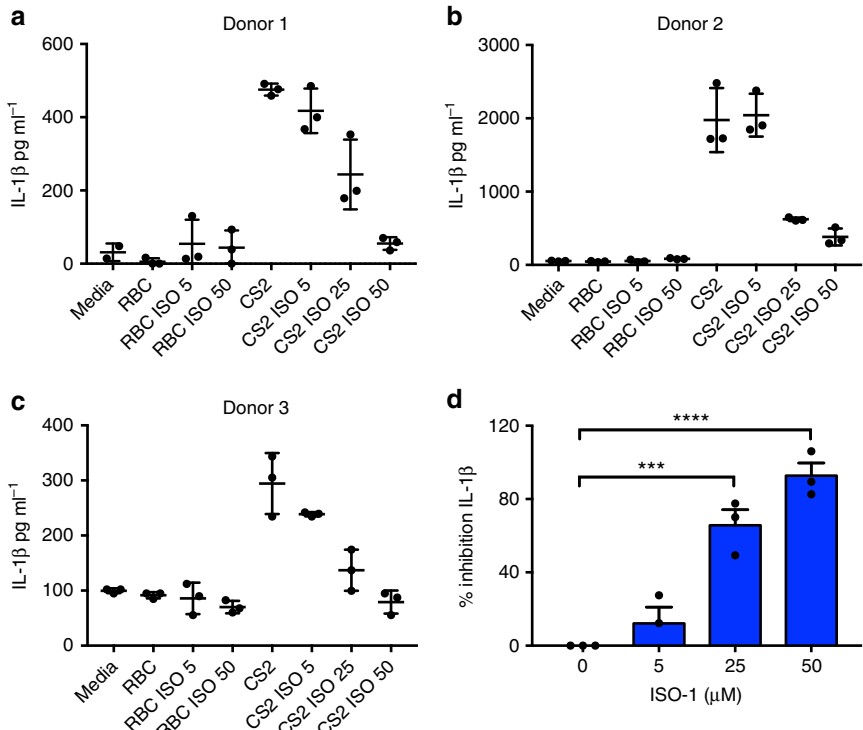

**Fig. 3** ISO-1 inhibits IL-1β in response to *Plasmodium falciparum*. **a–c** Viable human PBMCs collected from three donors were co-cultured with $2 \times 10^6$ non-infected or CS2-infected RBC. Cells were either treated with DMSO alone (control) or with increasing concentrations of ISO-1 (5, 25, or 50 μM) for 16 h. Levels of IL-1β in cell culture supernatants were quantified by ELISA. **d** Percentage inhibition calculated for all three donors ± SEM ***$P < 0.005$, ****$P < 0.001$, one-way ANOVA with a correction provided by the Tukey's multiple comparisons test

**Fig. 2** MIF regulates NLRP3-dependent release of IL-1 family cytokines. **a** Primary murine WT or $Mif^{-/-}$ BMDMs were left untreated, primed with LPS alone (10 ng ml$^{-1}$), or primed with LPS before transfection of poly dA:dT (1 μg ml$^{-1}$) for 5 h. WT BMDMs were left untreated, primed with LPS (10 ng ml$^{-1}$), or primed with LPS before the addition of COR123625 (50 μM) for 2 h before transfection of **b** poly dA:dT (1 μg ml$^{-1}$) or **c** flagellin (250 ng ml$^{-1}$) for 5 h. Alternatively, following priming of WT BMDMs with LPS, cells were treated with **d, g** MSU (150 μg ml$^{-1}$), **e, h** silica (150 μg ml$^{-1}$), or (**f, i**) PBI-F2 peptide (100 μg ml$^{-1}$) for 6 h. Levels of **a–f** IL-1β and **g–i** IL-18 in cell culture supernatants were quantified by ELISA. **j** Primary WT or $Mif^{-/-}$ BMDMs were primed with LPS (10 ng ml$^{-1}$) in the presence or absence of recombinant murine MIF (rMIF, 10 ng ml$^{-1}$) before stimulation with nigericin (5 μM) for 1 h. In addition, cells were treated with or without COR123625. Levels of IL-1β in cell culture supernatants were assessed by ELISA. **k** Primary WT or $Mif^{-/-}$ BMDMs were primed with LPS (10 ng ml$^{-1}$) in the presence or absence of MIF-containing conditioned media from WT BMDM before stimulation with nigericin (5 μM) for 1 h. Levels of IL-1β in cell culture supernatants were assessed by ELISA. Data are expressed as means ± SEM, $n = 3$ mice. *$P < 0.05$, **$P < 0.01$, ***$P < 0.005$, ****$P < 0.001$, one-way ANOVA with a correction provided by the Tukey's multiple comparisons test

malaria-endemic areas of Papua New Guinea with *Plasmodium falciparum* CS2-infected red blood cells (CS2-RBC) ex vivo, in the presence or absence of a MIF inhibitor. In three separate donors, we found that ISO-1 significantly inhibited CS2-RBC-induced IL-1β release by PBMC (Fig. 3a–d). Importantly, ISO-1 did not affect PBMC viability or phagocytic uptake of infected RBCs (Supplementary Figure 6A and 6B). These data further suggest that IL-1β release in response to *P. falciparum* is dependent on NLRP3 and MIF.

**MIF is required for NLRP3 inflammasome assembly**. The fact that inhibiting MIF specifically abrogates NLRP3-dependent IL-1β and IL-18 release suggests that MIF is involved in NLRP3 inflammasome activation or assembly, rather than directly influencing IL-1β production. To confirm this, we first investigated if LPS-dependent NF-κB activation was affected by inhibition of MIF, as it is required for the transcription of both IL-1β and NLRP3[23]. Murine macrophage RAW-ELAM cells, which stably express the ELAM-NF-κB reporter construct, were treated with LPS is the presence or absence of COR123625 or ISO-1. Inhibition of MIF had no effect on NF-κB activation in these cells (Fig. 4a). Next, we assessed *Il1b* mRNA expression in BMDM and the human THP-1 monocytic cell line. In both cell types, MIF inhibition did not inhibit LPS-induced expression of *Il1b* and, in fact, significantly increased LPS-induced *Il1b* expression in BMDM (Fig. 4b, c).

Next, we tested whether inhibition of MIF affected translation and processing of proteins pivotal for NLRP3 inflammasome assembly and activation. MIF inhibition with COR123625 had no effect on intracellular levels of pro-IL-1β, NLRP3, pro-caspase-1, or ASC protein in LPS-treated BMDM (Fig. 4d, e). However, COR123625 did inhibit the processing and release of both caspase-1 and IL-1β (Fig. 4d). Similarly, levels of intracellular pro-IL-1β in LPS-treated WT and *Mif*[−/−] BMDM were comparable (Supplementary Figure 7A and 7B). Moreover, COR inhibited IL-1β processing regardless of whether it was added before or after priming with LPS (Supplementary Figure 7C). Together, these data demonstrate that blocking MIF specifically inhibits NLRP3 inflammasome-dependent processing of pro-IL-1β, rather than LPS-dependent effects on protein expression.

Activation of the NLRP3 inflammasome is characterized by the formation of an ASC "speck"; the result of oligomerization of ASC molecules into a large protein complex[45]. Typically, in macrophages, this is followed by caspase-1-dependent cell death, or pyroptosis. To determine if MIF is required for inflammasome assembly and activation, we next evaluated the effects of COR123625 on NLRP3-mediated ASC-speck formation. In response to priming with LPS and NLRP3 activation with nigericin or silica, immortalized BMDM stably expressing fluorescent ASC-cerulean displayed significant ASC-speck formation over time, which was significantly inhibited when cells where treated with COR123625 (Fig. 5a–d and Supplementary Figure 8A and 8B). To test whether MIF inhibition also inhibited pyroptosis, we looked at cell death in BMDM following activation of NLRP3 with nigericin using a lactate dehydrogenase (LDH) release assay. As expected, treatment with LPS + nigericin significantly increased pyroptosis, while treatment with COR123625 abrogated this effect (Fig. 5e). Collectively, these data demonstrate that MIF is specifically required for initiation and assembly of the NLRP3 inflammasome upstream of ASC oligomerization.

Intriguingly, cleavage of caspase-1 to the p20 form was either no different, or higher in *Mif*[−/−] BMDM treated with LPS and nigericin compared to WT controls, and the *Mif*[−/−] cells also showed higher basal levels of pro-caspase-1 (Fig. 5f), potentially

suggesting a role for MIF in caspase-1 expression and/or degradation. Intracellular levels of pro-IL-1β (Supplementary Figure 7A and 7B) were comparable in WT and *Mif*[−/−] cells. Nonetheless, processing and release of IL-1β was still reduced compared to WT controls (Fig. 5f) and ASC-speck formation was reduced in *Mif*[−/−] BMDM (Fig. 5g, h). Thus, these data further confirm a role for MIF in ASC-speck/inflammasome formation.

**MIF mediates interactions between vimentin and NLRP3**. A recent study has demonstrated that activation and assembly of the NLRP3 inflammasome is dependent on interaction between NLRP3 and the intermediate filament protein vimentin[31]. In agreement with this, we found that treatment of macrophages with the vimentin-binding anti-angiogenic compound withaferin A abrogated ASC-speck formation in macrophages (Supplementary Figure 8C and 8D), while super-resolution microscopy revealed NLRP3 staining on vimentin-positive intermediate filaments in BMDM treated with LPS and nigericin (Supplementary Figure 8E). Thus, we hypothesized that MIF might be involved in mediating the NLRP3–vimentin interaction. To test this, we first performed co-immunoprecipitation (co-IP) experiments in WT and *Mif*[−/−] BMDM. In WT BMDM, vimentin could be detected in all samples following immunoprecipitation with NLRP3 and this interaction increased with LPS and nigericin (Fig. 6a). In *Mif*[−/−] BMDM, this interaction was greatly reduced (Fig. 6a).

However, because vimentin, which is a common contaminant in affinity purification mass spectrometry (Crapome.org, 2013; http://www.crapome.org/[46]), was also observed in immunoglobulin G (IgG) control samples, we sought to confirm these results using fluorescence lifetime imaging microscopy-fluorescent resonance energy transfer (FLIM-FRET), which only occurs when molecules are in close proximity (2–10 nm)[47]. In agreement with our co-IP data, increased FRET occurred between NLRP3 and vimentin in BMDM treated with LPS followed by nigericin, indicating increased interaction between the two molecules (Fig. 6b–d and Supplementary Figure 8F). This interaction was significantly reduced in the presence of COR123625 (Fig. 6b–d and Supplementary Figure 8F). Taken together, our data suggest that MIF is required for the interaction between NLRP3 and vimentin, which is in turn required for NLRP3 inflammasome assembly.

**MIF interacts with NLRP3**. Our data suggest a critical role for MIF in assembly of the NLRP3 inflammasome, facilitating interactions between NLRP3 and vimentin/intermediate filaments. Thus, we next looked to see whether MIF interacts with the NLRP3 inflammasome. Confocal analysis of MIF staining in ASC-cerulean immortalized BMDM (iBMDM) revealed co-localization of MIF and ASC on the ASC-speck (Fig. 7a). This was evident at early time points (5–20 min), but not later (30 min +), suggesting a role for MIF early in inflammasome assembly. Similarly, MIF could be seen to co-localize with NLRP3 after treatment of LPS-primed BMDM with nigericin (Fig. 7b). After 20 min, NLRP3 and MIF could be seen in large structures, similar to ASC specks. However, these structures were not evident in cells treated with nigericin and COR123625 (Fig. 7b).

We next sought to determine whether MIF interacts with NLRP3. Co-IP of iBMDM lysates, following treatment of LPS-primed cells with nigericin ± COR123625, using both MIF and NLRP3 antibodies, revealed direct interactions between MIF and NLRP3, as well as confirming the NLRP3–vimentin interaction (Fig. 8a). Similar to findings with vimentin and NLRP3, the interaction between MIF-NLRP3 was reduced in cells treated with COR12365. To test this further, we used FLIM-FRET analysis, which showed increased FRET between MIF and NLRP3 in BMDM treated with LPS and nigericin (Fig. 8b). However, pre-

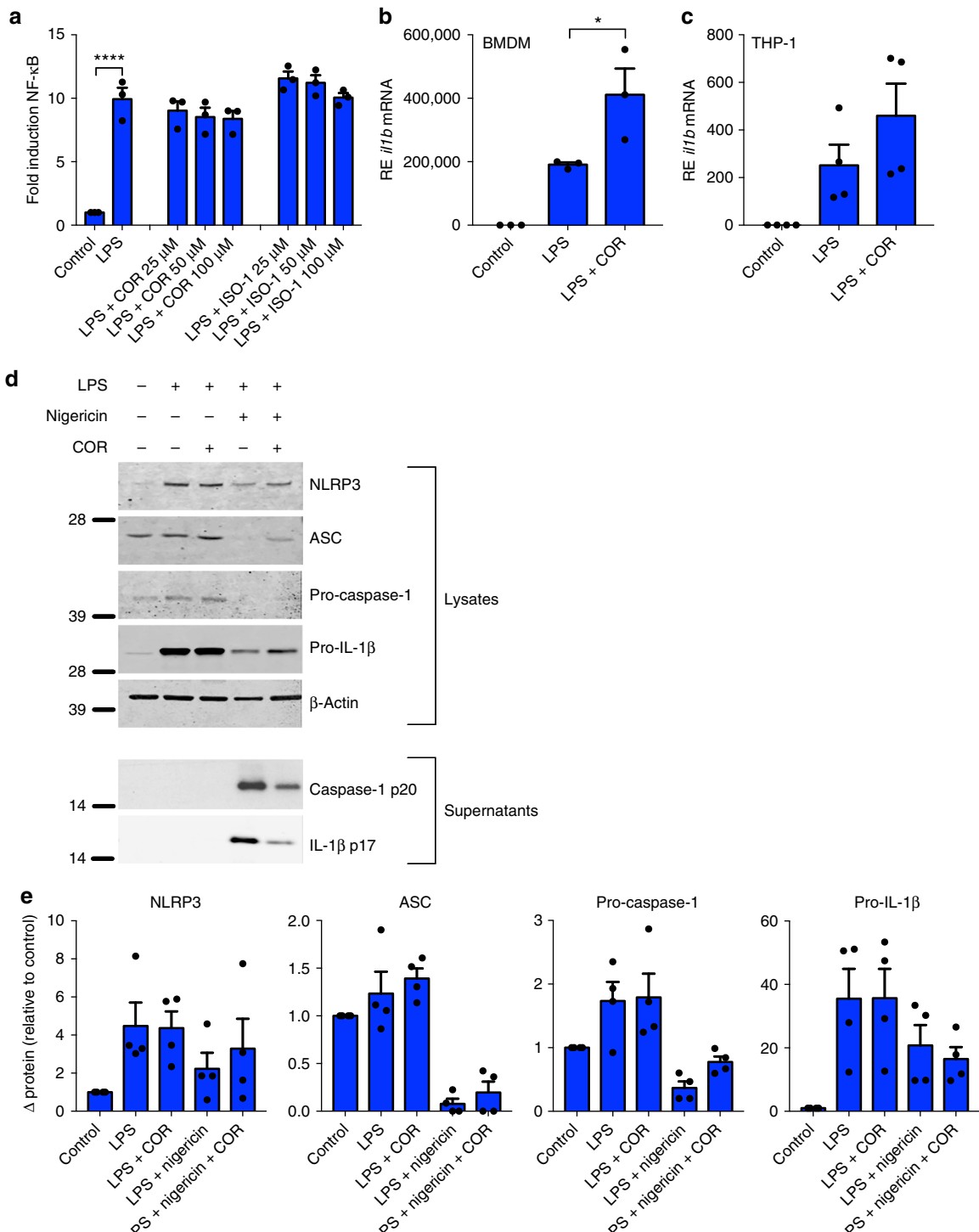

**Fig. 4** Inhibition of MIF does not prevent transcription or translation of IL-1β. **a** NF-κB luciferase activity measured in RAW-ELAM macrophages pre-treated with COR123625 (50 μM) for 1 h prior to priming with LPS (100 ng ml$^{-1}$) for 4 h. **b** Primary murine WT BMDM or **c** human undifferentiated THP-1 cells were left untreated, primed with LPS (100 ng ml$^{-1}$) alone for 4 h, or pre-treated with COR123625 (50 μM) for 1 h before the addition of LPS. Relative expression (RE—relative to 18S) of il1b mRNA was quantified using real-time PCR. Data are mean ± SEM, $n = 3$ independent experiments. *$P < 0.05$, ****$P < 0.001$, one-way ANOVA with a correction provided by the Tukey's multiple comparisons test. **d** WT BMDMs were left untreated, primed with LPS alone (100 ng ml$^{-1}$) for 5 h, pre-treated with COR123625 for 1 h prior to the addition of LPS, primed with LPS before inflammasome activation with nigericin (5 μM) for 1 h, or treated with COR123625 before LPS and nigericin stimulation. Western blot analysis of cellular supernatants and lysates to assess levels of NLRP3, ASC, pro-IL-1β, pro-caspase-1, mature IL-1β (p17), caspase-1 (p20), and β-actin was performed. **e** Densitometry was used to calculate expression of intracellular proteins shown in **d**. Mean expression was normalized to β-actin and expressed as relative to levels in control (untreated) samples $n = 4$ separate experiments (four mice)

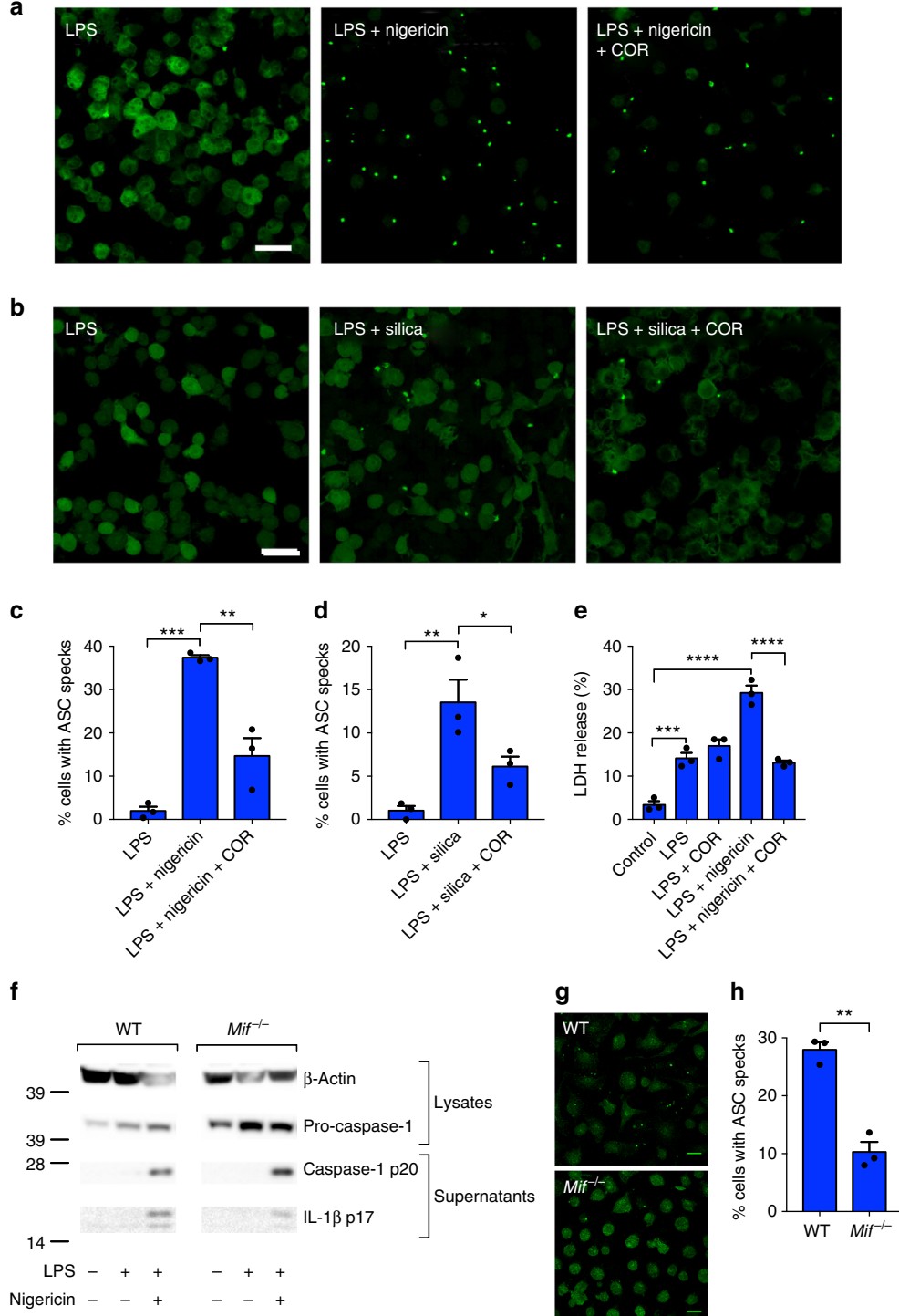

**Fig. 5** Inhibition of MIF prevents NLRP3 inflammasome activation. ASC-cerulean macrophages were primed with LPS (10 ng ml$^{-1}$) overnight. The following day cells were treated with COR123625 (50 μM) for 2 h before activation of the inflammasome with **a** nigericin (10 μM) (1 h) or **b** silica (150 μg ml$^{-1}$) (4 h). Confocal images are representative of three independent experiments. **c**, **d** Data are presented as the percentage of ASC-speck-positive cells. Data shown are mean ± SEM of three independent experiments. **e** WT BMDMs were left untreated, treated with LPS alone (100 ng ml$^{-1}$) for 5 h, treated with COR123625 for 1 h prior to the addition of LPS, primed with LPS before inflammasome activation with nigericin (5 μM) for 1 h, or treated with COR123625 before LPS and nigericin stimulation. Levels of LDH release were quantified using the Promega cytotoxicity assay. **f** WT or *Mif*$^{-/-}$ BMDMs were treated with LPS (10 ng ml$^{-1}$) + nigericin (5 μM) and lysates and supernatants analyzed by Western blot for caspase-1 and IL-1β. Images are representative of >3 mice. **g** BMDMs from WT and *Mif*$^{-/-}$ mice were treated with (10 ng ml$^{-1}$) + nigericin (5 μM), fixed and stained for ASC, and analyzed by confocal microscopy. Images are z projections of multiple z-stacks. **h** Quantitation of ASC specks in **g**, *n* = 3 mice per group. Data are expressed as percentage increase of mean ± SEM from three mice. *$P < 0.05$, **$P < 0.01$, ***$P < 0.005$, or ****$P < 0.001$, one-way ANOVA with a correction provided by the Tukey's multiple comparisons test

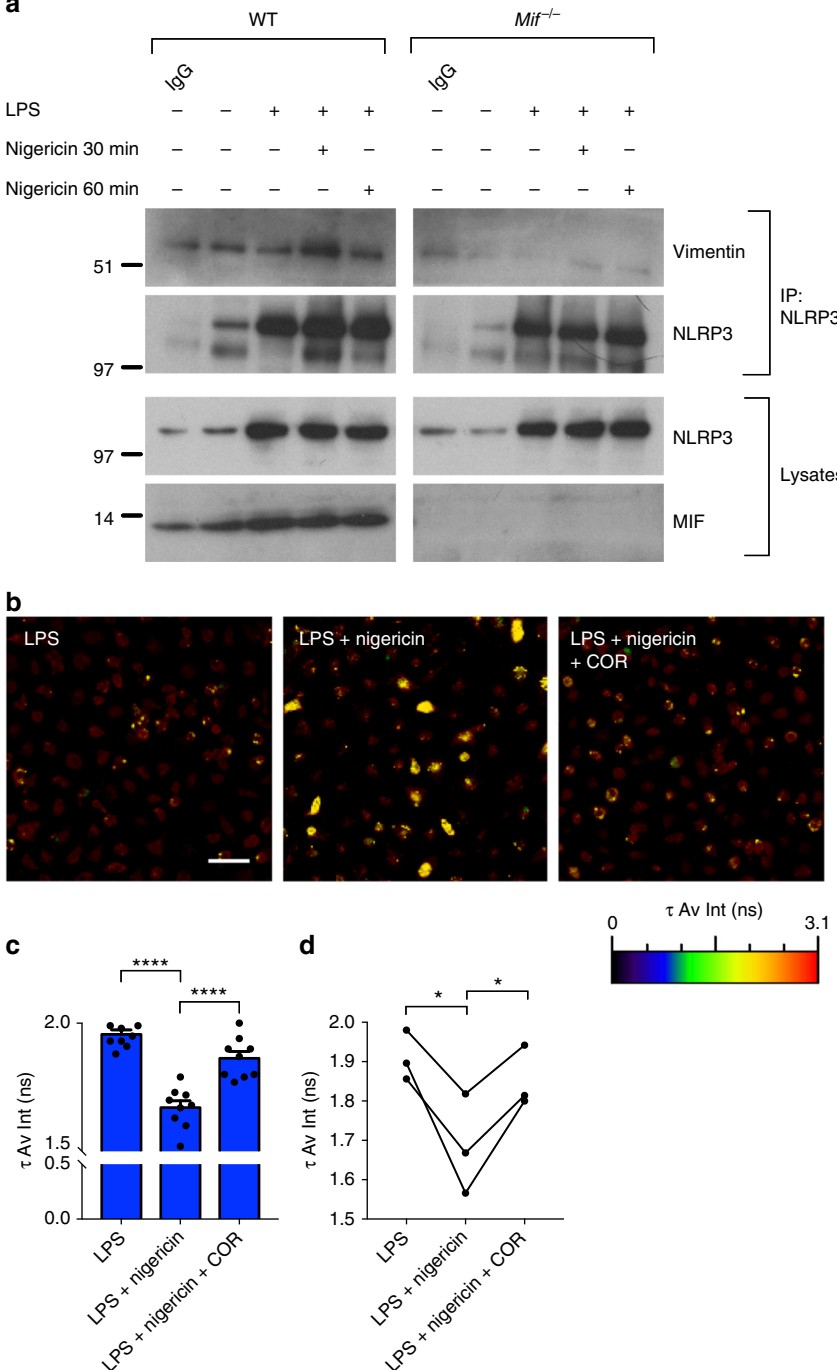

**Fig. 6** MIF is required for interactions between NLRP3 and vimentin. **a** Primary murine WT and $Mif^{-/-}$ BMDMs were left untreated, primed with LPS alone (100 ng ml$^{-1}$) for 5 h, or primed with LPS before inflammasome activation with nigericin (5 μM) for indicated times. Lysates were immunoprecipitated with anti-NLRP3 antibody, followed by Western blot analysis with anti-vimentin antibody. **b** WT BMDMs were primed with LPS alone (100 ng ml$^{-1}$) for 5 h, primed with LPS followed by nigericin (5 μM) for 1 h, or primed with LPS followed by COR123625 (50 μM) for 2 h prior to nigericin (5 μM) treatment for 30 min. Levels of interaction between NLRP3 and vimentin were assessed by FLIM-FRET. FLIM-FRET images presented are representative of three independent experiments (three mice). Scale bar = 50 μm. **c** Changes in the amplitude-weighted average lifetime ($\tau$ Av Amp) of the donor (A488) due to proximity to the acceptor (A568). Representative graph from one mouse (nine separate fields per mouse). ****$P < 0.001$, one-way ANOVA with a correction provided by the Tukey's multiple comparisons test. **d** Combined data from three mice, analyzed by paired $t$ test. *$P < 0.05$

treatment of the cells with COR123625 significantly inhibited this interaction (Fig. 8b). These data demonstrate, for the first time, that MIF interacts with NLRP3 and suggests that this interaction may be important for the interaction between NLRP3 and vimentin, which is required for assembly of the inflammasome.

## Discussion

MIF was one of the first cytokines identified and has been shown to have multiple roles in the immune system. In particular, it has a protective effect against a number of intracellular pathogens, but a pathological role in inflammatory and autoimmune diseases[14]. A likely common factor in both of these is the regulation

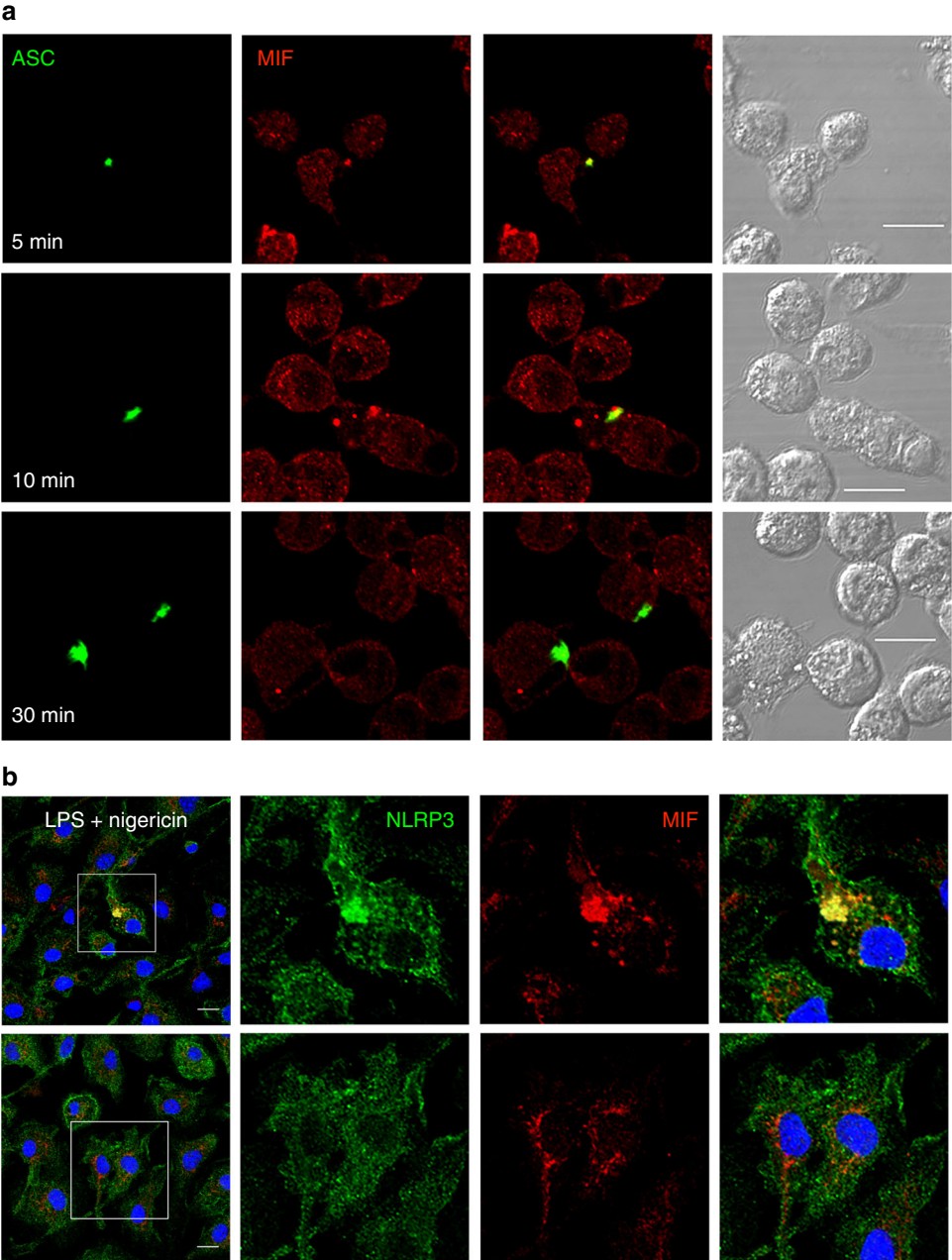

**Fig. 7** MIF co-localizes with the NLRP3 inflammasome. **a** ASC-cerulean macrophages were primed with LPS (10 ng ml$^{-1}$) overnight. The following day cells were treated with nigericin (10 μM) for the indicated times. Co-localization of ASC (green) and MIF (red) were visualized using confocal microscopy. Confocal images are representative of at least three independent experiments. **b** Primary murine BMDMs were treated with LPS (10 ng ml$^{-1}$) overnight, and then treated with or without COR123625 (50 μM), followed by nigericin (5 μM) for 20 min. Cells were fixed and stained with antibodies against NLRP3 and MIF. Nuclei were stained with DAPI. All scale bars = 10 μm

of other cytokines, increasing the expression and secretion of proinflammatory cytokines, while inhibiting anti-inflammatory molecules such as IL-10[38,48]. In addition, MIF has been shown to counteract the immunoregulatory effects of glucocorticoids (GCs)[49,50], making it an attractive therapeutic target, with the potential to both directly modify inflammation and improve sensitivity to GCs. Here, we have uncovered a previously unrecognized and highly specific role for MIF in the activation of the NLRP3 inflammasome and subsequent processing and release of the IL-1 family cytokines IL-1α, IL-1β, and IL-18. Previous work has demonstrated that loss of MIF can inhibit the LPS-induced release of TNF-α by downregulating TLR4 expression[51]. Although we did not look at TLR4 expression here, we saw no

effect of MIF inhibitors on TLR4-induced NF-κB activation and intracellular levels of pro-IL-1β were not affected by loss or inhibition of MIF. Moreover, similar effects of MIF inhibition on IL-1 release were seen when other TLR ligands were used. Interestingly, inhibition of MIF with COR123625 actually increased expression of *Il1b* mRNA. The reason for this is not clear, although previous studies have demonstrated similar dissociation between *il1b* mRNA and intracellular levels of protein[52,53]. It may be that mRNA stability is changed in COR-treated cells, or that post-translational regulation of pro-IL-1β is somehow altered (i.e., increased degradation). Given that loss of MIF blocks inflammasome activation and processing of pro-IL-1β, it may be that cells engage compensatory mechanisms to

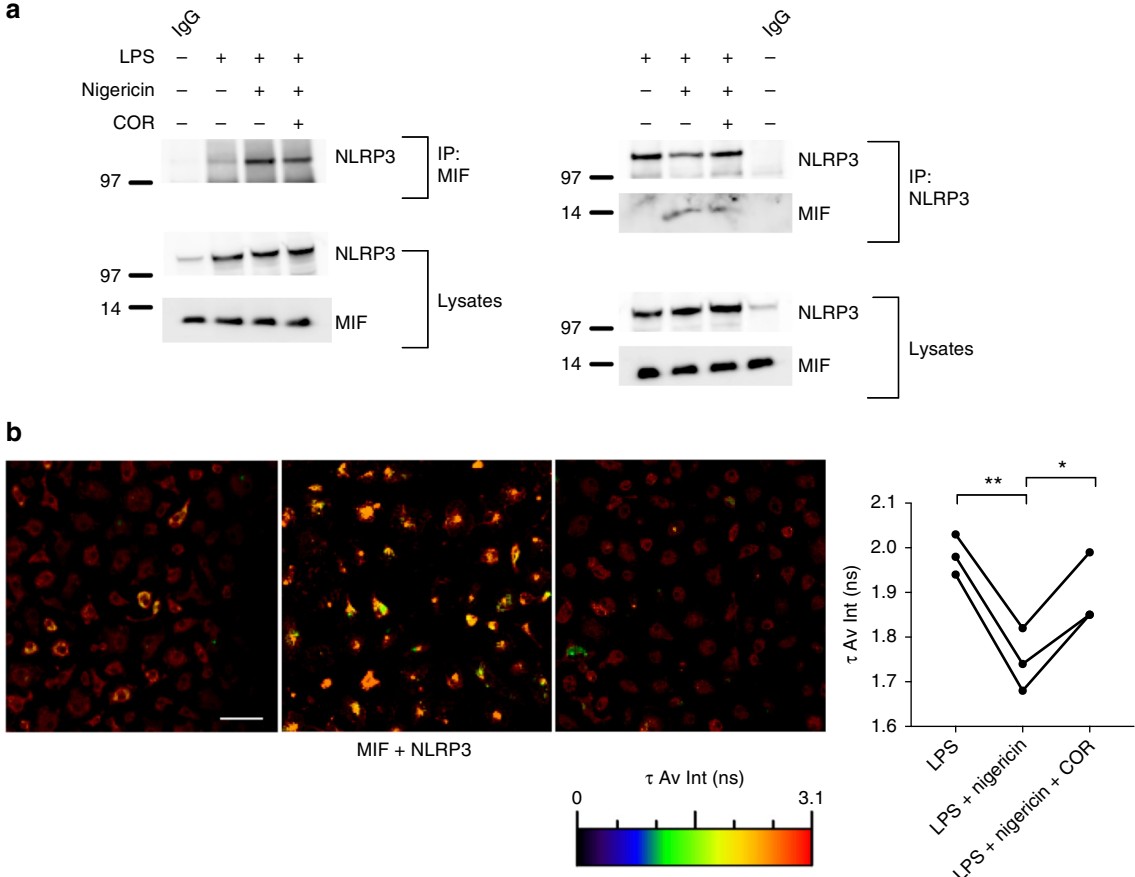

**Fig. 8** MIF interacts with NLRP3. **a** Immortalized BMDMs were left untreated, primed with LPS alone (100 ng ml$^{-1}$) for 5 h, or primed with LPS before inflammasome activation with nigericin (5 μM, 30 min). Lysates were immunoprecipitated with anti-MIF or anti-NLRP3 antibody, followed by western blot analysis with antibodies against NLRP3 and MIF. **b** WT BMDMs were primed with LPS alone (10 ng ml$^{-1}$) for 5 h, primed with LPS followed by NLRP3 activation with nigericin (5 μM) for 30 min, or primed with LPS followed by COR123625 (50 μM) treatment for 2 h prior to the addition of nigericin (5 μM) for 30 min. Levels of interaction between MIF and NLRP3 were assessed by FLIM-FRET microscopy. FLIM-FRET images presented are representative of three independent experiments. Scale bar = 50 μm. Graphs show changes in the amplitude-weighted average lifetime (τ Av Amp) of the donor (A488) due to proximity to the acceptor (A568). Combined data from three mice, analyzed by paired *t* test. \*$P < 0.05$, \*\*$P < 0.01$

increase IL-1β transcription, but why this does not equate to greater levels of protein is not clear.

The NLRP3 inflammasome has important roles to play in immune responses to multiple endogenous and exogenous insults, including MSU crystals, mitochondrial products, intracellular pathogens, and particulate material, such as silica and asbestos[62]. While much is known about stimuli that trigger NLRP3 activation, the process by which it is activated and assembled are less well understood. However, a recent study demonstrated a key role for the intermediate filament protein vimentin, which interacts with NLRP3 upon stimulation of cells with LPS and ATP, potentially providing a scaffold for subsequent activation and/or assembly[31]. Here, we demonstrate not only a novel role for MIF in regulating the NLRP3–vimentin interaction following activation but also that MIF itself interacts with NLRP3. Moreover, the dynamics of this interaction mirror those of NLRP3–vimentin, increasing after inflammasome activation and inhibited by blocking MIF. Precisely how MIF regulates NLRP3 activation is unclear at this stage. However, a number of studies have demonstrated that MIF has chaperone-like activity with model proteins[63]. More recently, a study has shown MIF to act as an ATP-independent protein folding chaperone, preventing misfolding of mutant superoxide dismutase[64], the accumulation of which accompanies development of the neurodegenerative disease amyotrophic lateral sclerosis (ALS).

Thus, it is possible that MIF might act as a chaperone for NLRP3, perhaps stabilizing the protein and/or promoting its interaction with vimentin. Further studies, beyond the scope of this paper, are required to uncover the precise mechanism through which MIF facilitates NLRP3 activation and assembly and how it binds NLRP3. Moreover, further studies to determine whether MIF also regulates NLRP3 oligomerization or the interactions between NLRP3 and ASC or NLRP3 and NEK7[32–34] would be of considerable interest. Importantly, however, our data suggest that this role of MIF is independent of its role as a cytokine, as recombinant and native MIF were unable to recover NLRP3-dependent IL-1 release in *Mif*$^{-/-}$ macrophages.

While inflammasome activation and caspase-1 are not exclusively required for the processing and release of IL-1α, NLRP3-activating stimuli are known to induce IL-1α release. In some cases, this may be the result of passive release from dying cells[55,56], but in others may be mediated via a caspase-1-dependent mechanism, albeit one that is independent of caspase-1 catalytic activity[56]. Correspondingly, our data demonstrate that loss or inhibition of MIF significantly inhibits IL-1α release in response to NLRP3-activating stimuli. Moreover, as with IL-1β, IL-1α release was not inhibited by MIF depletion or inhibition in response to the AIM2 and NLRC4 activators poly(dA:dT) and flagellin, respectively, suggesting that the inhibitory effects on IL-1α are NLRP3-dependent.

Intriguingly, our data suggest differences between acute inhibition of MIF with antagonists versus chronic MIF deficiency with respect to caspase-1. In particular, while the MIF inhibitor COR123625 clearly inhibited caspase-1 cleavage in macrophages, $Mif^{-/-}$ BMDM showed no discernable difference in caspase-1 p20 following treatment with the NLRP3 activator nigericin. Nonetheless, IL-1β processing and release was still inhibited in $Mif^{-/-}$ BMDM, possibly suggesting that the caspase-1 is either not activated, as has recently been suggested for p20[54], or is unable to cleave IL-1β in these cells. Moreover, our data demonstrate that ASC-speck formation is also inhibited in $Mif^{-/-}$ BMDM, further indicating that MIF is required for activation/assembly of the NLRP3 inflammasome.

Despite this difference with regard to caspase-1 p20, our data demonstrate that levels of pro-IL-1β are comparable in $Mif^{-/-}$ and WT BMDM and that it is specifically NLRP3-dependent IL-1β release that is inhibited in $Mif^{-/-}$ cells, suggesting that upstream effects on LPS signaling or downstream effects on IL-1β release are less likely to be responsible for the effects in $Mif^{-/-}$ cells. However, we cannot rule out the possibility that other, as yet undiscovered, effects of MIF depletion are simultaneously at play, potentially including effects on secretion/release. Importantly, COR123625 had no further effect on IL-1 release by $Mif^{-/-}$ cells either in vitro or in vivo, so it is unlikely that the inhibitor regulates NLRP3 in an MIF-independent manner. The increase in basal levels of pro-caspase-1 in $Mif^{-/-}$ BMDM could suggest a role for MIF in regulating production and/or degradation of the protein, or may be indicative of compensatory mechanisms at play in the knockout cells. Alternatively, as opposed to differences between chronic and acute loss of MIF, these results might instead suggest that MIF inhibitors that target the catalytic pocket of MIF (e.g., COR123625, ISO-1, 4-IPP) do not target other effects of MIF on IL-1 release pathways. However, further work is needed to fully understand the role of MIF in caspase-1 expression and activation, as well as the differences between MIF knockout and MIF antagonists with respect to caspase-1 activation.

Our data indicate a role for MIF in regulating IL-1β in human PBMCs in response to *P. falciparum*-infected erythrocytes. Previous studies have highlighted a role for IL-1β release in both protective and pathogenic responses to malaria[28,29,41,57,58]. In particular, IL-1 is thought to play a pivotal role in low birth weight due to placental malaria as it can decrease the activity of system A amino acid transporters in vitro and because placental IL-1 levels are negatively associated with birth weight in women suffering placental malaria with intervillositis[42,43]. Moreover, Plasmodium-induced IL-1β release has been shown to be dependent on both NLRP3 and AIM2[28,29,41]. Given the specificity for NLRP3 demonstrated here, our data suggest that, ex vivo, IL-1β release by PBMCs in response to *P. falciparum* infection is largely, if not entirely, NLRP3-dependent. Studies have also shown that MIF is increased in the placenta of malaria-infected women and is thought to be produced by leukocytes in the intervillous space[44,59]. Whether this increase in MIF is directly related to the increase in IL-1β is not clear and warrants further investigation. Interestingly *P. falciparum* produces a MIF homolog which can bind the MIF receptor CD74[60] and circulating levels of this Plasmodium-derived MIF (pMIF) are associated with parasitemia and a greater proinflammatory state in the host[60,61]. The precise role of this parasite-derived MIF is unclear, although in a mouse model it was shown to induce proinflammatory responses and dampen memory T cell responses by increasing terminal differentiation, which increases susceptibility to apoptosis, potentially reducing protection against subsequent infection[60]. Whether pMIF can regulate NLRP3-dependent responses remains to be seen, although our data with

native and recombinant MIF would suggest that exogenous MIF does not act on NLRP3 inflammasome activation.

An intriguing question that arises from our findings is how much this role in NLRP3 activation accounts for other known effects of MIF in the immune system. A recent study has demonstrated that MIF is required for IL-1β production and neutrophil accumulation in the joints of mice injected with MSU in a model of gout[36]. Our data provide a direct mechanism for this observation, particularly given that gout is an NLRP3-dependent pathology. Similarly, both MIF and NLRP3 have been shown to have pathological roles to play in mouse models of septic shock[38–40] and our findings suggest these effects may be interdependent. Moreover, other known effects of MIF may also be NLRP3-dependent, such as the reported protective roles for MIF in many microbial infections. For example, both MIF and NLRP3 have been shown to have a protective role in host immune responses to the fungal pathogen *Aspergillus fumigatus*[12,65]. Finally, a recent study has also suggested that NLRP3 upregulation is linked to GC resistance in leukemia cells[66], potentially indicating that MIF effects on NLRP3 activation might be behind its negative effects on GC sensitivity. Further work focusing on the role of this MIF–NLRP3 interaction in other known effects of MIF, including macrophage motility and recruitment and counteraction of GC activity, would be of considerable interest, both for our understanding of the biology of this molecule and for the development of future therapeutic strategies.

In summary, our data highlight a previously unrecognized role for MIF in NLRP3 inflammasome-dependent processing and release of IL-1 family cytokines. Depletion or inhibition of MIF in macrophages and dendritic cells resulted in the inhibition of IL-1α, IL-1β and IL-18 in response to NLRP3-activating stimuli. We demonstrate that loss of MIF does not alter the production of IL-1, but instead MIF is required for NLRP3–vimentin interactions, which are crucial for assembly of the inflammasome. Moreover, MIF interacts with NLRP3, potentially suggesting that MIF is directly involved in assembly and activation of the NLRP3 inflammasome.

## Methods

**Animals.** Eight-week-old male C57BL/6 mice (WT) and $Mif^{-/-}$ (on a C57BL/6 background)[67] mice were used with the approval of the appropriate Monash University Ethics Committee (MMCB) and in accordance with National and institutional guidelines. The $Mif^{-/-}$ mice were originally obtained from Professor R. Bucala, Yale School of Medicine. All animals were housed in individually ventilated cages, used at 8–12 weeks and culled using $CO_2$ asphyxiation.

**Cell culture.** Bone marrow cells were collected from femurs of WT and $Mif^{-/-}$ mice and differentiated for 7 days in RPMI 1640 (Thermo Fisher Scientific) supplemented with 10% heat-inactivated fetal calf serum (FCS) (JRH Biosciences, Victoria, Australia), 2 mM L-glutamine, 1% penicillin/streptomycin (Complete RPMI) and with 15% L929 conditioned medium. Differentiated BMDMs were plated on day 7 for treatment the following day. For BMDCs, bone marrow cells were differentiated for 11 days in complete RPMI with 10% granulocyte–macrophage colony-stimulating factor conditioned medium (Ag8653 myeloma cells). Differentiated primary BMDCs were plated on day 11 for treatment the following day. iBMDMs and ASC-cerulean-iBMDM were grown in Dulbecco's modified Eagle's medium supplemented with 10% FCS and 2 mM glutamine. The human monocytic THP-1 cell line was cultured in complete RPMI. RAW264.7 mouse macrophage cells stably expressing the NF-κB-dependent ELAM-luciferase reporter construct were cultured in complete RPMI with 0.5 mg ml$^{-1}$ G418. Cells were maintained at 37 °C in a humidified atmosphere of 5% $CO_2$.

**Collection of PBMC.** PBMCs, isolated from venous blood of three pregnant women at first antenatal visit, were collected in 2007 at Alexishafen Health Centre, Madang, Papua New Guinea, following written, informed consent. PBMCs were stored in liquid nitrogen until analysis. Viability and number of thawed PBMCs was assessed by trypan blue staining. Ethics approval was granted for this work by the Medical Research Advisory Council of PNG (05/08) and by the Melbourne Health Human Research Ethics Committee (2001.018.).

**In vivo LPS challenge**. Mice were injected intraperitoneally with 20 mg kg$^{-1}$ COR123625 or vehicle control (saline) in combination with 2 mg kg$^{-1}$ LPS or phosphate-buffered saline (PBS). After 2 h mice were killed, and serum levels of IL-1β, IL-18, and IL-6 were measured by ELISA.

**Inflammasome activation**. BMDM, iBMDM ($5 \times 10^5$ cells ml$^{-1}$), and THP-1 cells ($1 \times 10^6$ cells ml$^{-1}$) were plated in 96-well plates for cytokine and LDH quantification. The following day WT BMDMs were treated with 10 or 100 ng ml$^{-1}$ LPS, 10 μg ml$^{-1}$ imiquimod, or 10 ng ml$^{-1}$ Pam$_3$CSK$_4$ (all from Sigma) for 3–5 h or overnight. COR123625 (25–100 μM), ISO-1 (Abcam) (25–100 μM), and 4-IPP (Sigma) (50 μM) were added to cells either 1 h prior to LPS or 2 h prior to inflammasome activation. Cells were then stimulated with inflammasome activators: nigericin (Cayman Chemical) (5 or 10 μM for 5–60 min), adenosine 5′-triphosphate disodium salt (ATP, Sigma) (5 or 10 mM for 1 h), silica (Invivogen) (150 μg ml$^{-1}$ for 1–6 h), MSU (Invivogen) (150 μg ml$^{-1}$ for 6 h), PBI-F2 peptide (provided by A. Mansell) (100 μg ml$^{-1}$ for 6 h). Poly(dA:dT) (Sigma) (1 μg ml$^{-1}$ for 5 h) or flagellin (Sigma) (250 ng ml$^{-1}$ for 5 h) were transfected into cells with Lipofectamine 2000® reagent (Thermo Fisher Scientific). Supernatants were removed and analyzed using ELISA Kits according to the manufacturer's instructions (BioLegend). LDH release was measured using the CytoTox96® non-radioactive cytotoxicity assay (Promega).

**Infection of PBMC with *P. falciparum*-infected erythrocytes**. Type O erythrocytes (Australian Red Cross Blood Service) infected with *P. falciparum* strain CS2, which expresses the gene (*var2csa*) and phenotype (chondroitin sulfate-A binding) of parasites associated with placental malaria[68], were cultured in RPMI-HEPES supplemented with 0.2% NaHCO₃, 0.5% albumax II (Gibco), and 5% heat-inactivated pooled human serum (Australian Red Cross Blood Service) at 37 °C in a low oxygen environment (1% O₂, 5% CO₂, 94% N₂). Trophozoite stage infected erythrocytes were purified by Percoll gradient centrifugation prior to use as stimuli in the experiment. Non-infected RBCs (from the same donor as used for parasite culture) were used as a control.

A total of $2 \times 10^5$ viable PBMCs were co-cultured with $2 \times 10^6$ infected RBCs or non-infected RBC control/well (ratio 1:10) in complete RPMI with varying concentrations of ISO-1 or DMSO control for 16 h at 37 °C at 5% CO₂. Levels of IL-1β in cell culture supernatants were quantified using the human IL-1β ELISA Development Kit (MabTECH). Remaining cells in the wells were re-suspended in 5 μl of PBS, pipetted onto glass slides, then stained using Diff-Quik® (Baxter, Australia), and finally investigated by light microscopy to confirm phagocytosis of parasite by white blood cells.

**RNA isolation and quantitative real-time PCR**. BMDM or THP-1 cells were treated with LPS (100 ng ml$^{-1}$) for 4 h with or without COR123625 (50 μM). Total RNA was extracted using chloroform/isopropanol precipitation and complementary DNA (cDNA) was prepared from 0.5 μg of total RNA according to the M-MLV reverse transcription protocol using the Tetro cDNA Synthesis Kit (Bioline, Australia). Quantitative real-time PCR analyses were performed on duplicate samples using the Applied Biosystems Power SYBR Green PCR Master Mix (Thermo Fisher Scientific). PCR amplification was performed on the Rotor-Gene 3000 (Corbett Research, Australia). Relative gene expression was normalized to 18S.

**NF-κB luciferase reporter assay**. RAW-ELAM macrophages were pre-treated with COR123625 (50 μM) for 1 h prior to stimulation with LPS (100 ng ml$^{-1}$) for 4 h. Cells were lysed in 50 μl 1× passive Lysis Reporter Buffer (Promega). Luciferase activity was measured immediately after the addition of 30 μl of Luciferase Substrate Solution (Luciferase Assay System, Promega) using the Infinite M1000 Pro plate-reader (Tecan, Australia).

**Western blot**. Cells were lysed in KalB lysis buffer (50 mM Tris, 150 mM NaCl, 1% Trition X-100 and 1 mM EDTA, 1% NaF, 1% NaVO₄, 1 mM PMSF, and protease inhibitor cocktail, pH 7.4). Proteins from cell supernatants were extracted by methanol–chloroform precipitation. Protein samples were resolved on 15% NuPAGE® Bis-Tris gels (Novex, Thermo Fisher Scientific) and then transferred onto PVDF membrane. Membranes were blocked in 5% (w/v) milk powder in Tris-buffered saline with Tween-20 (TBS-T) (50 mM Tris-HCl, pH 7.6, 150 mM NaCl, and 0.1% (v/v) Tween-20) for 1 h at room temperature. Membranes were incubated with primary antibodies diluted in 5% (w/v) milk powder in TBS-T overnight at 4 °C and then with horseradish peroxidise-conjugated secondary antibody diluted in 5% (w/v) milk powder in TBS-T for 1 h at room temperature. Membranes were developed with Amersham ECL Western blotting detection reagents (GE Healthcare, Buckinghamshire, UK) and images either taken on film or using a Fujifilm LAS-4000 luminescent reader. Primary antibodies used were mouse anti-caspase-1 (AG-20B-0042, Adipogen), rabbit anti-IL-1β (ab9722, Abcam), rabbit anti-NLRP3 (D4D8T, Cell Signaling Technology), rabbit anti-MIF (ab7207, Abcam), rabbit anti-ASC (sc-22514-R, Santa Cruz), and mouse anti-vimentin (sc-373717, Santa Cruz).

**Co-immunoprecipitation**. BMDM lysates were prepared as for Western blotting. Lysates were then pre-cleared with 100 μl protein G sepahrose 4 fast flow beads (GE Healthcare) for 1 h at 4 °C. Beads were removed and lysates incubated with selected antibodies overnight at 4 °C. Fifty microliters of bovine serum albumin-blocked protein G separose beads were incubated with lysates for 2 h at 4 °C. Beads were washed in KalB lysis buffer, then boiled in 30 μl of Laemelli sample buffer for 5 min at 95 °C, and finally analyzed by Western blotting. Antibodies used were as for Western blot. In addition, normal rabbit IgG (sc-2027, Santa Cruz) was used for controls.

**Microscopy**. For confocal and stimulated emission depletion (STED) super-resolution microscopy, cells were cultured on glass coverslips and fixed with either ice-cold methanol for 5 min or 2% paraformaldehyde for 30 min. Before staining, cells were permeabilized with 0.1% Triton X-100 and blocked with 0.5% teleostein gelatin, 1% casein, and 5% goat serum in PBS. Cells were stained with primary antibody in block buffer for 1 h at room temperature, followed by fluorescently-conjugated secondary antibodies in PBS for 1 h at room temperature. Nuclei were stained with DAPI (4′,6-diamidino-2-phenylindole; 1 μg ml$^{-1}$, Sigma) for 5 min at room temperature. Cells were mounted in fluorescent mounting medium (Dako) and imaged on an Olympus FV1200 confocal microscope. Images were analyzed using the Imaris and Fiji imaging software.

For ASC-speck assays, either iBMDM stably expressing cerulean-ASC or primary BMDM stained with rabbit anti-ASC (sc-22514-R, Santa Cruz) were analyzed. Multiple z-sections were taken for each field of view to ensure all specks were recorded.

For FLIM-FRET, cells were cultured and stained as above. FLIM data was recorded using an Olympus FV1000 microscope equipped with a PicoHarp300 FLIM extension and a 485 nm pulsed laser diode from PicoQuant. Pixel integration time for FLIM images was kept at 40 μs per pixel and fluorescence lifetime histograms were accumulated to at least 10,000 counts in the maximum to ensure sufficient statistics for FLIM-FRET analysis. Photon count rates were kept at 5% of the laser repetition rate to prevent pileup. FLIM-FRET analysis was performed using the SymPhoTime 64 software (PicoQuant). Individual cells were chosen as regions of interest (ROIs), then the fluorescence lifetime decay of each ROI was deconvolved with the measured instrument response function and fitted with a biexponential decay. The amplitude-weighted average lifetime was extracted from each fit and averaged over all values of one sample condition. Primary antibodies used were as for Western blot (above), with the addition of biotinylated sheep anti-MIF (detection antibody from MIF DuoSet ELISA, part number 843857, R&D Systems). Secondary antibodies/reagents were anti-mouse-Alexa-Fluor 568, anti-rabbit-Alexa-Fluor 488, and streptavidin-Alexa-Fluor 568 (Abcam).

**Quantification and statistical analyses**. Data are presented as average values ± SEM from multiple individual experiments/animals or as average values ± SD from triplicate measurements in a representative experiment, as stated in figure legends. Statistical analysis was carried out using one-way analysis of variance (ANOVA), followed by Tukey's post hoc test, or paired Student's *t* test, using the GraphPad Prism software. *P* values <0.05 were considered significant.

**Data availability**. Data are available from the authors upon request.

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

## Acknowledgements

This work was supported by a Project Grant from the Australian Government National Health and Medical Research Council (NHMRC, APP1068040) and grants from the Rebecca L. Cooper Medical Research Foundation and the Lions Clubs of Victoria, Lions Clubs International Foundation and Australian Lions Foundation. In addition, this work was supported by the Victorian State Government Operational Infrastructure Scheme.

## Author contributions

T.L., E.H.A., C.S., M.C.J.W., A.M., E.F.M., and J.H. conceived and designed the experiments. T.L., J.P.W.L., K.E., S.J.C., A.A.P., H.F., N.S.D., D.A.K.T., E.H.A., I.M., D.S., F.S.B., C.S., M.D.T., and J.H. performed the experiments. T.L., K.E., E.H.A., and J.H. analyzed the data. T.L., E.F.M., and J.H. composed the manuscript.

## Additional information

**Competing interests:** The authors declare no competing interests.

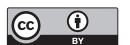

