## [Peer Review File · Nature Communications]

Reviewers' comments:

Reviewer #1 (Remarks to the Author):

This manuscript is an important addition to the NLRP3 inflammasome field. The authors describe a novel role of MIF and demonstrate that MIF is involved in NLRP3 inflammasome assembly and activity. Furthermore, MIF null cells and disruption of MIF activity using small molecule inhibitors demonstrate that NLRP3 activity is dependent on the presence of MIF. Overall the manuscript has novel insights to MIF and NLRP3 function. There are a few questions that need to be addressed and notes which better describe how the experiments were performed need to be added before the manuscript.

1. In Figure 4A, the authors report a fold-induction in NK-kB activation following treatment with LPS +/- COR. The activation levels are similar. The authors then examine the induction of IL1b mRNA in BMDM and THP-1 cells and report that cells treated with COR have a significantly elevated levels of IL1b mRNA. Please explain.
2. In Figure 4B and C, what is RE, relative expression?
3. The manuscript would be improved by quantifying the amount of each of the proteins in Figure 4D.
4. The data in figure 3 appears to be out of order in the manuscript and perhaps should be placed at the end of the manuscript. This would possibly provide a stronger clinical relevance for MIF inhibition and NLRP3 activity. Also, what was the rationale for switch between the MIF inhibitors? Do these inhibitors function identically in the two systems?
5. It's not clear why the authors designed a new MIF inhibitor, particularly if the existing inhibitors (ISO-1) were equally effective. It seems that 4-IPP or ISO-1, which are commercially available, could have been used instead.
6. In figure 6, there is significant concern regarding the co-immunoprecipitation experiments. The authors to fail to provide an input immunoblot which would demonstrate an equal amount of starting material. There appears to be decrease in the amount of NLRP3 expressed in MIF-/- cells at least by the amount of NLRP3 immunoprecipitated. Is this correct, in whole cell extracts is there a difference in NLRP3 expression?
7. The authors should have demonstrated that MIF is expressed in the protein complex containing vimentin, NLRP3 and MIF. Also, why is there protein expression in the IgG antibody control lane(s)? The IP should be conducted by using NLRP, MIF and vimentin and then immunoblotting for NLRP, MIF, and vimentin. As presented this data is incomplete.
8. The data in Figure 7 is not clear. What is the time course of ASC speck formation in the presence of LPS with or without COR? Why is the expression of the ASC speck formation altered after addition of COR? Can you prevent the immunoprecipitation of NLRP3/MIF and/or ASC/MIF in the presence of COR?
9. As ASC is supposed to assemble into a complex with NLRP3, is MIF still associated with the inflammasome complex when interacting with NLRP3? Does NLRP3 associate with the ASC speck at the various time points? What does an immunofluorescence image of MIF and NLRP3 look like? As figure 7B shows and illustrated in the graphs (7C and D) there is an interaction between NLRP3 and MIF. Is this interaction spread throughout the cell or focused at a single location, similar to the ASC speck? MIF appears to be distributed throughout the cell and NLRP3 is also or at least along vimentin filaments (see Supplemental data). How are these correlated?
10. A MIF speck is formed by 5 min that colocalizes with ASC after cells are treated with LPS and nigericin. What happens to MIF at various time points when COR is added? Does MIF associate with ASC? Does the ASC speck form? Is the assembly of the speck delayed? Answering these questions will help address how MIF interacts with the inflammasome. More importantly, a time course of MIF association with NLRP3, or a FLIM-FRET at different time points would address how COR affects the interaction between NLRP3 and MIF.
11. With regards to the FLIM-FRET assays, what fluorophores are being used? Where are the

proteins tagged? Do the modified proteins behave normally and are they localized correctly?
12. If MIF is added back to MIF KO BMDMs do the cells now behave as the controls?

Minor Comments

There are parts of the manuscript that need attention. For example what antibodies are used for western blots, immunofluorescence, immunoprecipitation. For the FLIM-FRET what vectors and constructs were used to express the proteins? At what time were the FLIM-FRET experiments performed?

Reviewer #2 (Remarks to the Author):

The authors provide evidence for a novel pathway by which MIF regulates IL-1 expression and inflammasome function. While some components of the data are supportive of such a pathway, relevant controls and comparator data are missing.

Specific Comments

Figure 1C is stated to show TNF levels in the Figure Legend, but is labeled IL-18.

Separate prior publications (Nature, PNAS) indicate a defect in macrophage cytokine production due to reduced LPS sensing via reduced TLR4 and to reduced increase activation induced apoptosis. Are these effects not operant in IL-1 expression in Fig 1 or in TNF production in Fig S1A?

Inhibition of IL-1B in conditioned medium by COR addition (small molecule MIF inhibitor) is shown. While dose-dependency is shown for catalytic activity, none is evident for biologic IL-1B release. Can the authors explain or demonstrate this ?

Fig 1G-H, and Fig 2 experiments. Given the distinction of COR action MIF with respect to inhibition of IL-1B versus TNF production in vitro (Fig S2E), it is puzzling that TNF data is not shown for these studies.

Fig 4D and Fig 5 – the MIF:COR inhibition studies should be complemented with controls with Mif^{-/-} cells.

Fig 6A – should include comparison with COR treatment.

Fig 7 - How was MIF labeled for confocal microscopy and for FRET analysis? The precise methodology is not trivial given possible induced perturbation in MIF functional structure.

Additional:

Several oversimplifications and misrepresentations are stated regarding the authors' description of MIF biology.

p. 4:

1. There are no experimental data this reviewer is aware of to indicate that MIF exists other than a homotrimer, so the statement "commonly" is misleading.
2. There is no structural homology between MIF and the stated enzymes; the claimed homology is strictly topologic (or by 3-dimensional structure) only.
3. The authors should refer to Fingerle et al (Mol Cell Biol) to better understand the relationship between cMIF catalytic activity and biologic function. Therein the authors dissociate these activities

by genetic knockin of a catalytically dead MIF to show elimination of catalytic activity and only partial attenuation of MIF receptor interaction.

4. It is an oversimplification to state that tautomerase inhibitors attenuate biologic activity; the vast majority of such compounds do not affect MIF biologic activity.

p. 8

What is a "knock-on" effect?

Reviewer #3 (Remarks to the Author):

In this manuscript, the authors present data showing that Macrophage migration inhibitory factor (MIF) regulates the activation of NLRP3 inflammasome. The authors show that MIF is required for the interaction between NLRP3 and the intermediate filament protein vimentin at the early stage of NLRP3 inflammasome activation. Then, the authors demonstrate that MIF interacts directly with NLRP3. This study is interesting, but substantial works need to be done to clarify the detailed mechanism.

Major concerns:

1. In Fig.1, it seems that COR123625 showed better inhibitory activity for nigericin-induced IL-1b production than deletion of MIF in BMDMs. They authors should examine whether this inhibitor could inhibit inflammasome activation in MIF KO cells.
2. To evaluate the role of inhibitor or MIF deletion on NLRP3 activation, the authors should examine the caspase-1 activation, not only IL-1 production.
3. In Fig.S1B, IL-6 production were reduced in MIF KO cells, suggesting that MIF might have some effects on the LPS-induced priming. The authors should carefully examine whether MIF deletion had effects on LPS-induced pro-IL-1b and NLRP3 expression.
4. In Fig.1G-I, the authors only evaluated the role of COR123625 on inflammation in WT mice, the effects of MIF-deletion should be examined. In addition, they should examine whether the inhibitor still had effects in MIF KO mice.
5. Although the authors claimed that MIF regulates NLRP3 inflammasome activation via interacting with NLRP3, they should examine whether MIF deletion had effects on the upstream signaling events of NLRP3, including potassium efflux, mitochondrial damage and NLRP3-NEK7 interaction.
6. For the interaction between NLRP3 and MIF, 1) they should examine the endogenous interaction during NLRP3 inflammasome activation by Co-IP and confocal/STED; 2) how MIF binds with NLRP3? which domains are responsible for the interaction?
7. The authors should study how MIF affected the interaction between NLRP3 and vimentin.
8. Does extracellular MIF contribute to NLRP3 inflammasome activation?

)Reviewer #1 (Remarks to the Author):

This manuscript is an important addition to the NLRP3 inflammasome field. The authors describe a novel role of MIF and demonstrate that MIF is involved in NLRP3 inflammasome assembly and activity. Furthermore, MIF null cells and disruption of MIF activity using small molecule inhibitors demonstrate that NLRP3 activity is dependent on the presence of MIF. Overall the manuscript has novel insights to MIF and NLRP3 function. There are a few questions that need to be addressed and notes which better describe how the experiments were performed need to be added before the manuscript.

1. In Figure 4A, the authors report a fold-induction in NK-kB activation following treatment with LPS +/- COR. The activation levels are similar. The authors then examine the induction of IL1b mRNA in BMDM and THP-1 cells and report that cells treated with COR have a significantly elevated levels of IL1b mRNA. Please explain.

Authors' response: This is a very good and interesting question, but not one we have an immediate answer to. What our data clearly shows is that although IL-1 β mRNA is increased, intracellular levels of pro-IL-1 β are not obviously affected. A previous study has demonstrated that cyclohexamide increases IL-1 β mRNA, but does not increase transcription (Turner *et al* 1989 – PMID:2511244), although the mechanism behind this is not clear. Other studies have presented similar results as well (Schindler *et al.* 1990 – PMID:2354999; Radwan *et al.* 2010 – PMID: 20713887 - and references therein). It may be that mRNA stability is changed in COR-treated cells, or that post-translational regulation of pro-IL-1 β is somehow altered (i.e. increased degradation). Given that loss of MIF blocks inflammasome activation and processing of pro-IL-1 β , it may be that cells engage compensatory mechanisms to increase IL-1 β transcription, but why this does not equate to greater levels of protein is not clear, but worthy of further investigation. We have amended the discussion to incorporate this.

2. In Figure 4B and C, what is RE, relative expression?

Authors' response: Relative expression (RE) *ilb* mRNA expression relative to 18S. This has been added to the figure legend.

3. The manuscript would be improved by quantifying the amount of each of the proteins in Figure 4D.

Author' response: We have quantified the proteins as suggested and included the data in Figure 4E.

4. The data in figure 3 appears to be out of order in the manuscript and perhaps should be placed at the end of the manuscript. This would possibly provide a stronger clinical relevance for MIF inhibition and NLRP3 activity. Also, what was the rationale for switch between the MIF inhibitors? Do these inhibitors function identically in the two systems?

Authors response: The decision to use ISO-1 here was a strategic one based on limited access to PBMC samples and not, at the time, having tested COR in human PBMC, whereas we knew that ISO-1 worked in human cells. We have since shown COR to work as well as ISO-1 in human THP-1 cells with regard to IL-1 β release to other

NLRP3 stimuli. It should be noted that we are not claiming COR to be a better compound than ISO-1. We would be happy to move this figure to the end if the editor agrees that we should.

5. It's not clear why the authors designed a new MIF inhibitor, particularly if the existing inhibitors (ISO-1) were equally effective..It seems that 4-IPP or ISO-1, which are commercially available, could have been used instead.

Authors' response: The new inhibitor was designed as part of a program initiated by E. Morand through a start-up company (Cortical) to develop new MIF inhibitors for potential therapeutic use in inflammatory diseases. COR123625 proved to be effective, although ultimately not potent enough for clinical use and similar in potency to ISO-1 and 4-IPP *in vitro*. *In vivo*, COR had favourable properties over ISO-1 we have extensive pharmacokinetic data and detailed binding data, and (importantly) we have a lot of it! Again, we are not proposing COR123625 as a superior inhibitor for these studies, or that it has clinical potential as it is, but as we have demonstrated, both ISO-1 and 4-IPP do have the same effect on IL-1 β production. Note, Cortical no longer exists and the IP for COR123625 has reverted to Monash University.

6. In figure 6, there is significant concern regarding the co-immunoprecipitation experiments. The authors to fail to provide an input immunoblot which would demonstrate an equal amount of starting material. There appears to be decrease in the amount of NLRP3 expressed in MIF^{-/-} cells at least by the amount of NLRP3 immunoprecipitated. Is this correct, in whole cell extracts is there a difference in NLRP3 expression?

Authors' response: We have updated this figure to include both NLRP3 and MIF in lysates from this experiment (Figure 6A). No differences were seen in levels of NLRP3 between WT and MIF^{-/-} cells. MIF staining was not seen in the KOs. Unfortunately, for technical reasons vimentin did not stain in these blots. However, we have added new co-IP data to Figure 8A which we hope helps to further address these concerns.

7. The authors should have demonstrated that MIF is expressed in the protein complex containing vimentin, NLRP3 and MIF. Also, why is there protein expression in the IgG antibody control lane(s)? The IP should be conducted by using NLRP, MIF and vimentin and then immunoblotting for NLRP, MIF, and vimentin. As presented this data is incomplete.

Authors' response: We have now run co-IPs using MIF and NLRP3 antibodies and staining for MIF, NLRP3 and vimentin (Figure 8A). We have not been able to get reliable pull down with our vimentin antibody to date. We commonly see vimentin in the IgG samples and vimentin is a common contaminant in these assays. See for example the CRAPome (<http://www.crapome.org/> - PMID: 23921808). A quick search of this shows that, for example, human vimentin is a contaminant in 257/411 control experiments. This is one of the reasons why we also used FLIM-FRET to confirm the observations around vimentin (FLIM-FRET also allows us to be more quantitative). We do not see similar contamination with the other proteins (MIF and NLRP3). Also, we have included data from FLIM-FRET on the MIF-vimentin interaction (Figure 8B), so while we do not have a vimentin pull down, we have instead shown interactions with MIF and NLRP3 pull downs, as well as with FLIM-FRET. We have commented on vimentin as a contaminant in the result section (Page 12).

8. The data in Figure 7 is not clear. What is the time course of ASC speck formation in the presence of LPS with or without COR? Why is the expression of the ASC speck formation altered after addition of COR? Can you prevent the immunoprecipitation of NLRP3/MIF and/or ASC/MIF in the presence of COR?

Authors' response: If this is referring to Figure S7 (A & B), ASC speck formation occurs in response to NLRP3 activation and so COR, which we have demonstrated inhibits NLRP3 activation, inhibits this process. Moreover, as the time course (with nigericin and silica) demonstrates, this is not just a change in the dynamics of speck formation (i.e. inhibiting MIF does not just slow down inflammasome activation). If the reviewer is referring to Figure 7A, COR was not used in this experiment and the time course was 5, 10 and 30 minutes. Using FLIM-FRET, we have shown that the MIF-NLRP3 is interrupted by COR (Figure 7B-D). FLIM-FRET data suggests that MIF and ASC do not interact (data not shown). Unless otherwise stated, FLIM-FRET experiments were conducted after 30 min treatment with nigericin (this has been clarified in the manuscript).

9. As ASC is supposed to assemble into a complex with NLRP3, is MIF still associated with the inflammasome complex when interacting with NLRP3? Does NLRP3 associate with the ASC speck at the various time points? What does an immunofluorescence image of MIF and NLRP3 look like? As figure 7B shows and illustrated in the graphs (7C and D) there is an interaction between NLRP3 and MIF. Is this interaction spread throughout the cell or focused at a single location, similar to the ASC speck? MIF appears to be distributed throughout the cell and NLRP3 is also or at least along vimentin filaments (see Supplemental data). How are these correlated?

Authors' response: We propose that MIF might be involved in the process by which NLRP3 and ASC associate to form the inflammasome and this is why we see MIF co-localising with the ASC speck, particularly at early time points (so yes, we think MIF is associated with the inflammasome). We also see NLRP3 (and caspase-1) co-localising with the ASC speck at these time points (data not shown), although both remain co-localised after 30 minutes, while most of the MIF co-localisation is gone at this point. The FLIM-FRET data show that the interaction between MIF and NLRP3 occurs throughout the cell and not only at a single location. We also have data (both co-IP and FLIM-FRET) to suggest that MIF interacts with vimentin, suggesting that MIF, NLRP3 and vimentin are closely associated. We have included this data in Figure 7.

10. A MIF speck is formed by 5 min that colocalizes with ASC after cells are treated with LPS and nigericin. What happens to MIF at various time points when COR is added? Does MIF associate with ASC? Does the ASC speck form? Is the assembly of the speck delayed? Answering these questions will help address how MIF interacts with the inflammasome. More importantly, a time course of MIF association with NLRP3, or a FLIM-FRET at different time points would address how COR affects the interaction between NLRP3 and MIF.

Authors' response: Our co-IP and FLIM-FRET data suggest that MIF does not interact with ASC. The MIF speck is, we propose, MIF co-localising with the ASC speck due to its interaction with NLRP3. In the presence of COR, ASC speck formation is

significantly inhibited, so fewer MIF “specks” are seen. From the confocal studies, there is no other clear difference in MIF distribution/localisation inside the cells. As stated above, COR inhibits ASC speck formation over time with both nigericin and silica (Figure S8A & S8B), suggesting that this not just a change in dynamics. We have also included a timecourse FLIM-FRET experiment looking at the MIF-NLRP3 interaction and this is shown in Figure S8F. The data shows the most significant effects on the interaction at 15 and 30 minutes. In addition, we have added confocal pictures of MIF and NLRP3 to Figure 7B

11. With regards to the FLIM-FRET assays, what fluorophores are being used? Where are the proteins tagged? Do the modified proteins behave normally and are they localized correctly?

Authors’ response: The FLIM-FRET studies were conducted using antibodies against the endogenous proteins (not modified proteins). We have updated the materials and methods to describe this in more detail. For all FLIM-FRET experiments, a combination of Alexa 488 and 568 tagged secondary antibodies was used. No modified proteins were used.

12. If MIF is added back to MIF KO BMDMs do the cells now behave as the controls?

Authors’ response: Importantly, No. We tested the effects of both recombinant MIF and MIF-containing media from MIF-competent BMM on MIF^{-/-} BMM and saw no effect on IL-1 β release in response to LPS and nigericin. We have now included the results of these experiment in Figure 2J and 2K. This suggests to us that the role of MIF in NLRP3 activation is independent of its function as a cytokine.

Minor Comments

There are parts of the manuscript that need attention. For example what antibodies are used for western blots, immunofluorescence, immunoprecipitation. For the FLIM-FRET what vectors and constructs were used to express the proteins? At what time were the FLIM-FRET experiments performed?

Authors’ response: Apologies for these omissions. These issues have now been addressed in the updated materials and methods. Note that the FLIM-FRET experiments were done using antibodies against the endogenous proteins on fixed cells. Timings for FLIM-FRET experiments have been added to the figure legends.

Reviewer #2 (Remarks to the Author):

The authors provide evidence for a novel pathway by which MIF regulates IL-1 expression and inflammasome function. While some components of the data are supportive of such a pathway, relevant controls and comparator data are missing.

Specific Comments

Figure 1C is stated to show TNF levels in the Figure Legend, but is labeled IL-18.

Authors' response: Apologies for this mistake. We have corrected the figure and text.

Separate prior publications (Nature, PNAS) indicate a defect in macrophage cytokine production due to reduced LPS sensing via reduced TLR4 and to reduced increase activation induced apoptosis. Are these effects not operant in IL-1 expression in Fig 1 or in TNF production in Fig S1A?

Authors' response: While we have not looked directly at TLR4 expression, we have demonstrated in some detail that MIF inhibition does not affect LPS-induced NF- κ B activation in these cells and does not affect levels of IL-1 β protein in response to LPS (Figure 4A,D,E, Figure S7A,B). Moreover, we see similar effects with other TLR ligands (imiquimod and PM₃CysK₄) (Figure S4) and still see inhibition of IL-1 β processing and release if MIF is inhibited *after* LPS stimulation (New Figure S7C). Combined with the effects we clearly see on NLRP3 activation, we therefore conclude that the effect of MIF inhibition on IL-1 β and IL-18 release is independent of TLR4 expression. Also, as demonstrated in Figure 5E, COR has no effect on cell death in response to LPS. This is consistent with other experiments we have conducted, suggesting that MIF inhibition does not alter cell viability under the conditions described in this manuscript (except for when pyroptosis is induced, as in Figure 5E).

Inhibition of IL-1B in conditioned medium by COR addition (small molecule MIF inhibitor) is shown. While dose-dependency is shown for catalytic activity, none is evident for biologic IL-1B release. Can the authors explain or demonstrate this?

Authors' response: In the experiments presented here we have not titrated COR down enough to see a full dose response (see, for example, Figure S2). It could be that tautomerase activity is irrelevant to the effects on NLRP3, or that the tautomerase assay and IL-1 assays are not directly comparable. However, note that in the experiments with ISO-1 and *P. falciparum*-infected erythrocytes, we do see a very clear dose response with ISO-1 (Fig. 3). (This may suggest that COR is effective at lower doses than ISO-1). Also, we have very clear dose response curves with ISO-1 in THP-1 cells treated with nigericin (see below).

Fig 1G-H, and Fig 2 experiments. Given the distinction of COR action MIF with respect to inhibition of IL-1 β versus TNF production *in vitro* (Fig S2E), it is puzzling that TNF data is not shown for these studies.

Authors' response: The focus of this study is on the role of MIF in the release of IL-1 family cytokines, rather than TNF (which has been published previously). For the experiment in Fig 1-G-I we looked at IL-6, because this was seen to be significantly inhibited *in vitro* (Figure S1B), whereas TNF was not (Figure S1A). Because we were using ELISA, rather than multiplex, we were limited by serum volume as to how many cytokines we could test. Again, we are not suggesting that COR does not effect TNF *in vivo* (and by means other than those at play for the effects on IL-1 family cytokines), but this has been published and is not the focus of this study. Similarly, we fully accept that the effects on IL-6 might be due to effects of MIF on other pathways and have acknowledged this in the text. This does not, however, change the primary observations reported here in terms of NLRP3 activation.

Fig 4D and Fig 5 – the MIF:COR inhibition studies should be complemented with controls with Mif^{-/-} cells.

Author's response: We have now done this and show the results in Figure 2J,K. COR had no effect on IL-1 β release in MIF^{-/-} cells *in vitro*. Moreover, COR had no effect on serum IL-1 β in MIF^{-/-} mice injected i.p. with LPS (Supplemental Figure S4F and S4G)

Fig 6A – should include comparison with COR treatment.

Authors' response: Effects of COR on co-IP is now shown in Figure 8A.

Fig 7 - How was MIF labeled for confocal microscopy and for FRET analysis? The precise methodology is not trivial given possible induced perturbation in MIF functional structure.

Authors' response: For FLIM-FRET, MIF was labelled after fixation with a biotinylated anti-MIF antibody and then secondary-labelled with streptavidin-A568. For confocal studies, a rabbit anti-MIF antibody was used. The materials and methods have been updated to include more detail on this. At no point was MIF labelled in live cells.

Additional:

Several oversimplifications and misrepresentations are stated regarding the authors' description of MIF biology.

p. 4:

1. There are no experimental data this reviewer is aware of to indicate that MIF exists other than a homotrimer, so the statement "commonly" is misleading.

Authors' response: We have changed this line to "exists as a homotrimer".

2. There is no structural homology between MIF and the stated enzymes; the claimed homology is strictly topologic (or by 3-dimensional structure) only.

Authors' response: This has been corrected in the text.

3. The authors should refer to Fingerle et al (Mol Cell Biol) to better understand the relationship between cMIF catalytic activity and biologic function. Therein the authors dissociate these activities by genetic knockin of a catalytically dead MIF to show elimination of catalytic activity and only partial attenuation of MIF receptor interaction.

Authors' response: We have added this to the introduction to address this important point.

4. It is an oversimplification to state that tautomerase inhibitors attenuate biologic activity; the vast majority of such compounds do not affect MIF biologic activity.

Authors' response: We have updated the manuscript to address these issues. It should be noted that we are not claiming here that tautomerase activity is required for the effects we have shown (we really don't know at this stage) and acknowledge that studies have demonstrated it is not required for all biological activity of MIF.

p. 8

What is a "knock-on" effect?

By knock-on effect, we are referring here to effects of IL-1 β regulation on the release of other cytokines and signalling pathways. For example, we and others have demonstrated that IL-1 β can drive the autocrine release of IL-23 and TNF- α through an NF- κ B dependent process (Harris *et al* 2008 – PMID: 18802048, Peral de Castro *et al* 2012 – PMID: 22972933).

Reviewer #3 (Remarks to the Author):

In this manuscript, the authors present data showing that Macrophage migration inhibitory factor (MIF) regulates the activation of NLRP3 inflammasome. The authors show that MIF is required for the interaction between NLRP3 and the intermediate filament protein vimentin at the early stage of NLRP3 inflammasome activation. Then, the authors demonstrate that MIF interacts directly with NLRP3. This study is interesting, but substantial works need to be done to clarify the detailed mechanism.

Major concerns:

1. In Fig.1, it seems that COR123625 showed better inhibitory activity for nigericin-induced IL-1b production than deletion of MIF in BMDMs. They authors should examine whether this inhibitor could inhibit inflammasome activation in MIF KO cells.

Authors' response: We have tested this and the answer is no, COR does not inhibit inflammasome activation in MIF^{-/-} BMM cells (figure 2J). It should be noted the degree to which both knockout and MIF inhibitors regulate NLRP3 activation (particularly IL-1 β release) does vary from experiment to experiment (although the effect is always the same). This is evident throughout our presented data – so sometimes the inhibitors appear more effective, other times the KOs give a clearer result.

2. To evaluate the role of inhibitor or MIF deletion on NLRP3 activation, the authors should examine the caspase-1 activation, not only IL-1 production.

Authors' response: We have demonstrated that caspase-1 activity (and IL-1 β processing) is inhibited by MIF inhibitors (Fig. 4D).

3. In Fig.S1B, IL-6 production were reduced in MIF KO cells, suggesting that MIF might have some effects on the LPS-induced priming. The authors should carefully examine whether MIF deletion had effects on LPS-induced pro-IL-1b and NLRP3 expression.

Authors' response: Levels of pro-IL-1 β were comparable in LPS-treated WT and MIF^{-/-} cells, suggesting no obvious effects on LPS-induced priming (data now added to Figure S7A,B). As shown in Fig. 4, COR had no effect on LPS-induced pro-IL-1 β expression (nor on levels of NLRP3, ASC or pro-caspase-1). Moreover, we see the same effect on IL-1 secretion in cells primed with imiquimod (TLR7) or Pam₃CysK₄ (TLR2) and also see inhibition of IL-1 β processing regardless of whether MIF inhibitors are added before or after LPS priming (now added to Figure S7C). Also, neither COR nor ISO-1 had any effect on LPS-induced NF- κ B activation in RAW cells (Figure 4A). While we do not rule out the possibility that effects on IL-6 are due to differences in LPS signaling in MIF^{-/-} cells, those potential differences are not obvious and do not appear to be relevant to IL-1 β processing and release, which is the primary focus of this study.

4. In Fig.1G-I, the authors only evaluated the role of COR123625 on inflammation in WT mice, the effects of MIF-deletion should be examined. In addition, they should examine whether the inhibitor still had effects in MIF KO mice.

Authors' response: We have also looked at i.p. LPS in MIF^{-/-} mice with COR and have added this to Supplemental Figure S4F, G. Serum IL-1 β was significantly lower in MIF^{-/-} mice, but COR had no further effect on levels. Similarly, COR did not affect serum IL-6 levels in MIF^{-/-} mice. As mentioned above, we have also shown that COR has no effect on nigericin-induced IL-1 β release by MIF^{-/-} BMDM *in vitro* (Figure 2J).

5. Although the authors claimed that MIF regulates NLRP3 inflammasome activation via interacting with NLRP3, they should examine whether MIF deletion had effects on the upstream signaling events of NLRP3, including potassium efflux, mitochondrial damage and NLRP3-NEK7 interaction.

Authors' response: Our data clearly demonstrates that MIF interacts with NLRP3 and vimentin (new data shown in Figure 8) and that inhibition of MIF interrupts these interactions, as well as the NLRP3-vimentin interaction. Thus, we feel that effects on upstream events are unlikely to be relevant, given these very clear interactions. We do have some data (which we are not including here as it is incomplete and we have had issues with reagents) to show that inhibiting MIF does interfere with the NLRP3-NEK7 interaction, but there is no clear evidence that this is upstream of the NLRP3-vimentin interaction (and is downstream of K⁺ efflux), but instead is likely to be part of the same process of inflammasome assembly. We have added a note in the discussion that it would be interesting to look at this in further studies. We have also tested to see whether MIF inhibition with COR affects the release of mitochondrial ROS following NLRP3 activation, but found that mtROS release was not actually increased in iBMDM treated with LPS/nigericin and so was not a pertinent factor in our studies (see figure here). This is consistent with our own previous findings that mitochondrial damage is more of an issue when autophagy is impaired or inhibited (Harris *et al* 2011 – PMID: 21228274, Lee *et al* 2016 – PMID: 27163877). (We also have data demonstrating that MIF inhibition does not change autophagy induction in these experiments). We have not included this negative data in the manuscript, but would be happy to do so if requested.

6. For the interaction between NLRP3 and MIF, 1) they should examine the endogenous interaction during NLRP3 inflammasome activation by Co-IP and confocal/STED; 2) how MIF binds with NLRP3? which domains are responsible for the interaction?

We have now demonstrated the interaction between endogenous MIF and NLRP3 (and MIF and vimentin) by co-IP, as well as by FLIM-FRET (Figure 8). How MIF binds NLRP3 is a very interesting question, but is beyond the scope of the present study. It would require considerable extra work and funding and delay publication of what we feel is an important and exciting discovery. If we get further funding, we will certainly be looking into this.

7. The authors should study how MIF affected the interaction between NLRP3 and vimentin.

Again, we feel this is beyond the scope of this study, requiring a significant body of extra work, but would certainly be of considerable interest in the future.

8. Does extracellular MIF contribute to NLRP3 inflammasome activation?

We have conducted experiments to test this and included the results in Figure 2. Neither recombinant MIF, nor media from MIF-competent cells was able to reconstitute NLRP3 activation in MIF^{-/-} macrophages. This data strongly suggests that this effect of MIF is not dependent on its role as a cytokine, but rather a cell-intrinsic effect.

Reviewers' comments:

Reviewer #1 (Remarks to the Author):

The author's responses to the reviewer's critiques and additions to the manuscript have improved the manuscript. The rewritten portions of the manuscript demonstrates MIF interactions with both NLRP3 and vimentin and demonstrates that there is a complex regulation between MIF, NLRP3, vimentin, inflammasome activation and cytokine release.

One thing the authors need to address is in figure 4E. The protein quantification from the western blots should be from more than a single set of blots. Additionally, separating the quantification into separate figures will make analysis of the figure easier. For example, the amount of ASC in the LPS + nigericin + COR is it 5% of the control 10% or 50%?

Reviewer #2 (Remarks to the Author):

The revised manuscript is improved in presentation and this reviewer's concerns have been satisfactorily addressed.

Reviewer #3 (Remarks to the Author):

In the revised manuscripts, the authors addressed several concerns, but there are several concerns remain:

1. The authors provided the data showing that the MIF inhibitor suppressed caspase-1 activation, but they should also examine whether caspase-1 activation was impaired in MIF KO cells.
2. Another concern is that the current data can not clarify how MIF-NLRP3-vimentin interaction regulates NLRP3 inflammasome assembly. The authors should examine 1) how MIF interacts with NLRP3, 2) whether MIF KO whether affected the upstream signalings, NLRP3-NEK7 interaction, NLRP3 oligimerization, NLRP3-ASC interaction or other signaling events which are essential for NLRP3 activation.

Response to Reviewer's Comments

All changes to the document have been highlighted.

Reviewer #1

The author's responses to the reviewer's critiques and additions to the manuscript have improved the manuscript. The rewritten portions of the manuscript demonstrates MIF interactions with both NLRP3 and vimentin and demonstrates that there is a complex regulation between MIF, NLRP3, vimentin, inflammasome activation and cytokine release.

One thing the authors need to address is in figure 4E. The protein quantification from the western blots should be from more than a single set of blots. Additionally, separating the quantification into separate figures will make analysis of the figure easier. For example, the amount of ASC in the LPS + nigericin + COR is it 5% of the control 10% or 50%?

Authors response: We have now separated the individual proteins and combined the data from 4 separate animals as requested. This is now shown in Figure 4D

Reviewer #2

The revised manuscript is improved in presentation and this reviewer's concerns have been satisfactorily addressed.

Reviewer #3

In the revised manuscripts, the authors addressed several concerns, but there are several concerns remain:

1. The authors provided the data showing that the MIF inhibitor suppressed caspase-1 activation, but they should also examine whether caspase-1 activation was impaired in MIF KO cells.

Authors response: We have now performed many experiments to address this comment. Surprisingly, we see either no difference in caspase-1 p20 in *Mif*^{-/-} BMDM compared to WT, or, in some cases, more (we have tested this is a total of 9 mice per group, with varying results for caspase-1). We have added a representative blot in Figure 5F, which shows increased basal levels of caspase-1 (which we saw in 8 out of 9 mice) and increased caspase-1 p20 (in 9 mice, we saw increased p20 in 4, decreased in 1 and similar levels in the remaining 4). However, processed IL-1 (p17) was reduced and indeed in all experiments throughout the paper, NLRP3-dependent IL-1 release was consistently reduced in MIF deficient cells. This would suggest that the caspase-1 is either less active, or is not in contact with the IL-1. Levels of intracellular pro-IL were no different in *Mif*^{-/-} cells and no processed

IL-1 was observed in lysates. Levels of NLRP3 were also comparable between WT and knockouts. To try and make at least some sense of this apparently confounding result, we also looked at ASC speck formation in WT and knockout cells and found that this was significantly lower in the Mif^{-/-} cells (Figure 5G,H). Thus, this result, along with others in the paper demonstrating that NLRP3-dependent IL-1 and the NLRP3-vimentin interaction are inhibited in knockouts, confirm our hypothesis that MIF is indeed required for NLRP3 activation/assembly. We have discussed possible reasons why the caspase-1 data is different in the Mif^{-/-} cells, including differences between knockouts and inhibitors and reference to an interesting new paper from Kate Schroder's group demonstrating that caspase-1 p20 is actually an inactive form of the enzyme. As with many aspects of this work, this observation would be worthy of further investigation, but it is currently outside the scope of this study.

2. Another concern is that the current data can not clarify how MIF-NLRP3-vimentin interaction regulates NLRP3 inflammasome assembly. The authors should examine 1) how MIF interacts with NLRP3, 2) whether MIF KO whether affected the upstream signalings, NLRP3-NEK7 interaction, NLRP3 oligomerization, NLRP3-ASC interaction or other signaling events which are essential for NLRP3 activation.

Authors response: *Again, we agree that these are interesting and important questions, but would require an extensive amount of additional work that we argue is outside the scope of what is already a large amount of data reporting a very novel and potentially very important finding with regard to MIF and NLRP3. We have noted this in our discussion and hope that, should this work be accepted, it will inspire other labs to address some of these questions (and potentially help us to get the interest and funding required to take this work further ourselves).*

Reviewers' Comments:

Reviewer #3:

Remarks to the Author:

It is strange that deletion of MIF suppresses ASC speckle formation and IL-1 secretion, but has no effects on caspase-1 cleavage. Since the major claim of this study is "MIF is required for NLRP3 inflammasome activation", the authors need to carefully evaluate why there are inconsistencies (MIF inhibitor VS MIF KO; IL-1 secretion VS Caspase-1 activation) . This is very important for looking for the real mechanisms for the impaired IL-1 β production in MIF KO cells.

Response to Reviewers comments:

It is strange that deletion of MIF suppresses ASC speckle formation and IL-1 secretion, but has no effects on caspase-1 cleavage. Since the major claim of this study is "MIF is required for NLRP3 inflammasome activation", the authors need to carefully evaluate why there are inconsistencies (MIF inhibitor VS MIF KO; IL-1 secretion VS Caspase-1 activation). This is very important for looking for the real mechanisms for the impaired IL-1 β production in MIF KO cells.

Author's Response:

We agree that the difference between *Mif*^{-/-} and inhibitors is strange and, to be honest, difficult to explain at this stage. We have added caveats to the revised discussion that the reason for this is unclear at this stage, but may reflect differences between acute (inhibitor) vs chronic (KO) loss of MIF, or that compensatory mechanisms are at play in the KO cells. It is certainly not unusual for KO animals to behave in strange and unexpected ways. Alternatively, it may be that MIF interacts with the IL-1 release pathway in more ways than one, but that only the effect on NLRP3 activation/interaction with vimentin is targeted by the inhibitors (all of which target the same site on the MIF trimer). Nonetheless, our data still confirm that ASC speck formation and IL-1 processing are affected in the KO cells. In the end, we have stated that "...*further work is needed to fully understand the role of MIF in caspase-1 expression and activation, as well as the differences between MIF knockout and MIF antagonists with respect to caspase-1 activation.*" However, and very importantly, these data do not change the fact that MIF antagonists do target NLRP3 activation/interaction with vimentin and that this is a MIF-specific effect (i.e. is not present in *Mif*^{-/-} cells).